# DISCRETE CODEBOOK WORLD MODELS FOR CONTINUOUS CONTROL

**Aidan Scannell**[†]
University of Edinburgh
aidan.scannell@ed.ac.uk

**Mohammadreza Nakhaei**[*]
Aalto University

**Kalle Kujanpää**[*]
Aalto University

**Yi Zhao**
Aalto University

**Kevin Sebastian Luck**
Vrije Universiteit Amsterdam

**Arno Solin**
Aalto University

**Joni Pajarinen**
Aalto University

## ABSTRACT

In reinforcement learning (RL), world models serve as internal simulators, enabling agents to predict environment dynamics and future outcomes in order to make informed decisions. While previous approaches leveraging discrete latent spaces, such as DreamerV3, have demonstrated strong performance in discrete action settings and visual control tasks, their comparative performance in state-based continuous control remains underexplored. In contrast, methods with continuous latent spaces, such as TD-MPC2, have shown notable success in state-based continuous control benchmarks. In this paper, we demonstrate that modeling discrete latent states has benefits over continuous latent states and that discrete codebook encodings are more effective representations for continuous control, compared to alternative encodings, such as one-hot and label-based encodings. Based on these insights, we introduce DCWM: **D**iscrete **C**odebook **W**orld **M**odel, a self-supervised world model with a discrete and stochastic latent space, where latent states are codes from a codebook. We combine DCWM with decision-time planning to get our model-based RL algorithm, named DC-MPC: **D**iscrete **C**odebook **M**odel **P**redictive **C**ontrol, which performs competitively against recent state-of-the-art algorithms, including TD-MPC2 and DreamerV3, on continuous control benchmarks. See our project website www.aidanscannell.com/dcmpc.

## 1 INTRODUCTION

In model-based reinforcement learning (RL), world models (Ha & Schmidhuber, 2018) have been introduced in order to simulate or predict the environment's dynamics in a data-driven way. An agent equipped with a world model can make predictions about its environment by "simulating" possible actions within the model and "imagining" the outcomes. This equips the agent with the ability to plan and anticipate outcomes given a (learned) reward function, and the additional ability to envision transitions and outcomes before taking them in the real world can in turn improve sample efficiency.

One of the state-of-the-art world models, DreamerV2/V3 (Hafner et al., 2022; 2023) achieves strong performance in a wide variety of tasks, by "imagining" sequences of future states within a world model and using them to improve their policies. Interestingly, DreamerV2/V3 introduced a discrete latent space, in the form of a one-hot encoding, which offered significant benefits over its predecessor, DreamerV1 (Hafner et al., 2019a). This suggests that discrete latent spaces may have benefits over continuous latent spaces. It could be from the discrete latent space helping avoid compounding errors over multi-step time horizons or enabling policy and value learning to harness the benefits of discrete variable processing for efficiency and interoperability. In the context of generative modeling, discrete codebooks have been at the heart of many successful approaches (Chang et al., 2023; Esser et al., 2021; Ramesh et al., 2021). However, in the context of continuous control, TD-MPC2 (Hansen et al., 2023) uses a continuous latent space and significantly outperforms DreamerV3. Whilst there are multiple differences between TD-MPC2 and DreamerV2/V3, in this paper, we are specifically interested in exploring if discrete latent spaces can offer benefits for continuous control.

---

[†]Work done while at Aalto University    [*]Equal contribution

Recently, Farebrother et al. (2024) showed that training value functions with classification may have benefits over training with regression. The benefits may arise because *(i)* classification considers uncertainty during training (via the cross-entropy loss), *(ii)* the categorical distribution is multi-modal so it can consider multiple modes during training, or *(iii)* learning in discrete spaces is more efficient. In the context of world models, it is natural to ask, what benefits are obtained by *(i)* using discrete vs continuous latent spaces and *(ii)* modeling deterministic vs stochastic transition dynamics. Further to this, when considering stochastic latent transition dynamics, what is the effect of modeling with *(i)* unimodal distributions (*e.g.* Gaussian in continuous latent spaces) vs *(ii)* multimodal distributions (*e.g.* categorical in discrete latent spaces). In this paper, we explore these ideas in the context of world models for model-based RL, *i.e.* does learning a discrete latent space using classification have benefits over learning a continuous latent space using regression.

**Contributions**  The main contributions are as follows:

(C1)  In the context of continuous control, we show that learning discrete latent spaces with classification does have benefits over learning continuous latent spaces with regression.
(C2)  We show that formulating a discrete latent state using codebook encodings has benefits over alternatives, such as one-hot (like DreamerV2/V3) and label encodings.
(C3)  Based on our insights, we introduce Discrete Codebook World Model (DCWM): a world model with a discrete latent space where each latent state is a discrete code from a codebook. It obtains strong performance in the difficult locomotion tasks from DeepMind Control suite (Tassa et al., 2018) and manipulation tasks from Meta-World (Yu et al., 2019).

## 2 RELATED WORK

In this section, we recap world models in the context of model-based RL. We introduce two competing methods for learning latent spaces *(i)* those using observation reconstruction and *(ii)* those using latent state temporal consistency objectives. We then compare methods that learn continuous latent spaces using regression and those that learn discrete latent spaces using classification.

**World models**  Model-based RL is often said to be more sample-efficient than model-free methods. This is because it learns a model in which it can reason about the world, instead of simply trying to learn a policy or a value function to maximize the return (Ha & Schmidhuber, 2018). The world model can be used for planning (Allen & Koomen, 1983; Basye et al., 1992). A prominent idea has been to optimize the evidence lower bound of observation and reward sequences to learn world models that operate on the latent space of a learned Variational Autoencoder (VAE, Kingma & Welling (2014); Igl et al. (2018)). These models rely on maximizing the conditional observation likelihood $p(o_t|z_t)$, *i.e.* the reconstruction objective. The latent space of the model can then be used for both policy learning in the imagination of the world model, known as offline planning, *e.g.* Dreamer (Hafner et al., 2019a), or for decision-time planning (Rubinstein, 1997; Hafner et al., 2019b; Schrittwieser et al., 2020).

**Latent-state consistency**  Using the reconstruction loss for learning latent state representations is unreliable (Lutter et al., 2021) and can have a detrimental effect on the performance of model-based methods in various benchmarks (Kostrikov et al., 2020; Yarats et al., 2021a). To this end, TD-MPC (Hansen et al., 2022) and its successor, TD-MPC2 (Hansen et al., 2023), use a consistency loss to learn representations for planning with Model Predictive Path Integral (MPPI) control together with reward and value functions learned through temporal difference methods (Williams et al., 2015). Note that many prior works learn latent state representations using variants of a self-supervised latent-state consistency objective (Schwarzer et al., 2020; Wang et al., 2022; Ghugare et al., 2022; LeCun; Georgiev et al., 2024; Scannell et al., 2024b; Zhao et al., 2023; Scannell et al., 2024a). Given the success of learning representations without observation reconstruction in continuous control tasks, we predominantly focus on this class of methods, *i.e.* methods that use latent-state consistency losses.

**Discrete latent spaces**  DreamerV1 (Hafner et al., 2019a), DreamerV2 (Hafner et al., 2022), and DreamerV3 (Hafner et al., 2023), are world model methods which learn policies using imagined transitions from their world models. They utilize observation reconstruction when learning their world models and perform well across a wide variety of tasks. However, they are significantly outperformed by TD-MPC2 in continuous control tasks, which does not reconstruct observations. Of particular interest in this paper, is that DreamerV2/V3 introduced a discrete latent space, in the form of a one-hot encoding, and trained it with a classification objective, significantly improving performance. In contrast,

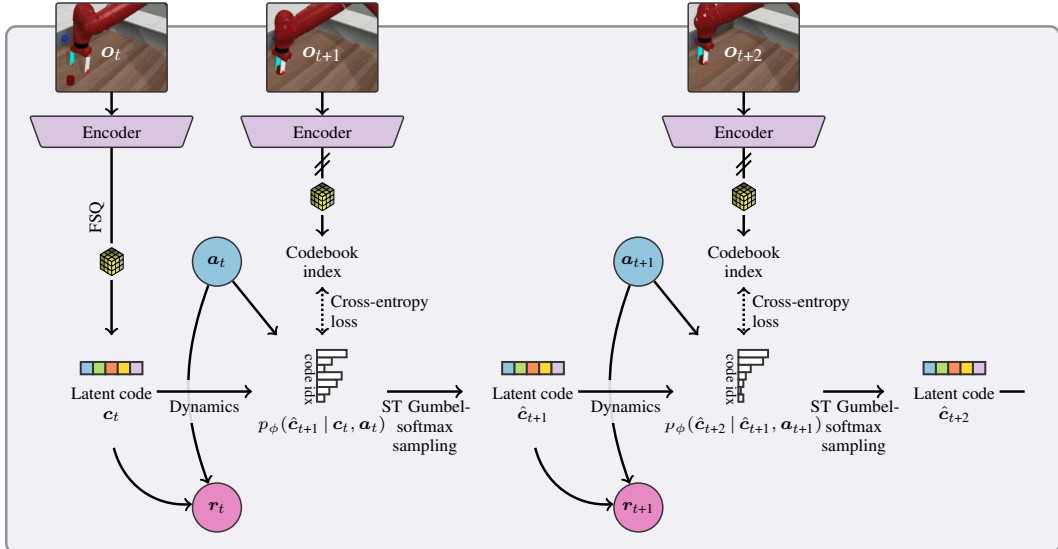

Figure 1: **World model training** DCWM is a world model with a discrete latent space where each latent state is a discrete code $c$ (⬛🟦⬛⬛🟥⬛) from a codebook $\mathcal{C}$. Observations $o$ are first mapped through the encoder and then quantized (⬢) into one of the discrete codes. We model probabilistic latent transition dynamics $p_\phi(c' \mid c, a)$ as a classifier such that it captures a potentially multimodal distribution over the next state $c'$ given the previous state $c$ and action $a$. During training, multi-step predictions are made using straight-through (ST) Gumbel-softmax sampling such that gradients backpropagate through time to the encoder. Given this discrete formulation, we train the latent space using a classification objective, *i.e.* cross-entropy loss. Making the latent representation stochastic and discrete with a codebook contributes to the very high sample efficiency of DC-MPC.

TD-MPC2 learns a continuous latent space with mean squared error regression. In this paper, we are interested in learning discrete latent spaces with classification, however, in contrast to DreamerV2/V3, we seek to avoid observation reconstruction – due to its poor performance in continuous control (see Fig. 20) – and instead learn the latent space using a self-supervised latent-state consistency loss.

## 3 PRELIMINARIES

In this section, we recap different types of discrete encodings and compare their pros and cons. First, let us assume we have three discrete categories: $A$, $B$, and $C$.

- **One-hot encoding** Given categories $A$, $B$, and $C$, a one-hot encoding would take the form $e(A) = [1, 0, 0]$, $e(B) = [0, 1, 0]$, and $e(C) = [0, 0, 1]$ respectively.
- **Label encoding** Given categories $A$, $B$, and $C$, label encoding would result in $e(A) = 1$, $e(B) = 2$, and $e(C) = 3$ respectively.
- **Codebook encoding** Given categories $A$, $B$, and $C$, a codebook might encode them as $e(A) = [-0.5, -0.5]$, $e(B) = [0, 0]$, and $e(C) = [0.5, 0.5]$ respectively.

**Ordinal relationships** If we have an ordinal relationship $A < B < C$, label and codebook encodings can ensure $|e(A) - e(B)| < |e(A) - e(C)|$, where $e(\cdot)$ is the encoding function. In this case, the global ordering is preserved along both dimensions of the codebook. It is worth noting that codebook encodings are flexible enough to model ordinal relationships in multiple dimensions. For example, the following code vectors exhibit opposite ordering along their two dimensions $e(E) = [0.5, -0.5]$, $e(F) = [0, 0]$, $e(G) = [-0.5, 0.5]$, which adds a level of modeling flexibility. One-hot encoding, however, results in $|e(A) - e(B)| = |e(A) - e(C)| = \sqrt{2}$ for all distinct pairs, eliminating any notion of ordering. Whilst this may be beneficial in some scenarios, *e.g.*, when modeling distinct categories like fruits, it means that they cannot capture the inherent ordering in continuous data.

**Sparsity and dimensionality** Another downside of one-hot encodings is that they create sparse data (*i.e.*, data with many zero values), which can have a negative impact on neural network training. In contrast, label and codebook encodings create dense data (*i.e.* many non-zero values). Finally, it is worth

noting that one-hot encodings have high dimensionality, especially when there are many categories. This makes them memory-intensive and slow to train when using a large number of categories.

In this work, we show that discrete codebook encodings resulting from quantization (Mentzer et al., 2024) offer benefits over both one-hot and label encodings when learning discrete latent spaces for continuous control. This is because they preserve ordinal relationships in multiple dimensions whilst being simpler, much lower-dimensional and having less memory requirements.

## 4 METHOD

In this section, we detail our method, named *Discrete Codebook Model Predictive Control* (DC-MPC), which is a model-based RL algorithm which *(i)* learns a world model with a discrete latent space, named *Discrete Codebook World Model* (DCWM), and then, *(ii)* performs decision-time planning with MPPI. The paper's main contribution is formulating a discrete latent space using quantization such that latent states are codes from a codebook. This allows us to train the latent representation using classification, in a self-supervised manner. See Fig. 1 for an overview of DCWM, Alg. 1 for details of world model training and Alg. 2 for details on the MPPI planning procedure.

We consider Markov Decision Processes (MDPs, Bellman (1957)) $\mathcal{M} = (\mathcal{O}, \mathcal{A}, \mathcal{P}, \mathcal{R}, \gamma)$, where agents receive observations $\boldsymbol{o}_t \in \mathcal{O}$ at time step $t$, perform actions $\boldsymbol{a}_t \in \mathcal{A}$, and then obtain the next observation $\boldsymbol{o}_{t+1} \sim \mathcal{P}(\cdot \mid \boldsymbol{o}_t, \boldsymbol{a}_t)$ and reward $r_t = \mathcal{R}(\boldsymbol{o}_t, \boldsymbol{a}_t)$. The discount factor is denoted $\gamma \in [0, 1)$.

### 4.1 WORLD MODEL

Learning world models with discrete latent spaces (*e.g.* DreamerV2) has proven powerful in a wide variety of domains. However, these approaches generally perform poorly in continuous control tasks when compared to algorithms like TD-MPC2 and TCRL (Zhao et al., 2023), which use continuous latent spaces. Rather than representing a discrete latent space using a one-hot encoding, as was done in DreamerV2, DC-MPC aims to construct a more expressive representation which is effective for continuous control. More specifically, DC-MPC represents discrete latent states as codes from a discrete codebook, obtained via finite scalar quantization (FSQ, Mentzer et al. (2024)). The world model can subsequently benefit from the advantages of discrete representations, *e.g.* efficiency and training with classification, whilst performing well in continuous control tasks.

**Components** DC-MPC has six main components:

$$\text{Encoder:} \quad \boldsymbol{x} = e_\theta(\boldsymbol{o}) \in \mathbb{R}^{|\mathcal{L}| \times d} \tag{1}$$

$$\text{Latent quantization:} \quad \boldsymbol{c} = f(\boldsymbol{x}) \in \mathcal{C} \tag{2}$$

$$\text{Dynamics:} \quad \boldsymbol{c}' \sim \text{Categorical}\left(p_1, \ldots, p_{|\mathcal{C}|}\right) \quad \text{with } p_i = p_\phi(\boldsymbol{c}' = \boldsymbol{c}^{(i)} \mid \boldsymbol{c}, \boldsymbol{a}) \tag{3}$$

$$\text{Reward:} \quad r = R_\xi(\boldsymbol{c}, \boldsymbol{a}) \in \mathbb{R} \tag{4}$$

$$\text{Value:} \quad \boldsymbol{q} = \mathbf{q}_\psi(\boldsymbol{c}, \boldsymbol{a}) \in \mathbb{R}^{N_q} \tag{5}$$

$$\text{Policy prior:} \quad \boldsymbol{a} = \pi_\eta(\boldsymbol{c}) \tag{6}$$

The encoder $e_\theta(\cdot)$ first maps observations $\boldsymbol{o}$ to continuous latent vectors $\boldsymbol{x} \in \mathbb{R}^{b \times d}$, where the number of channels $b$ and the latent dimension $d$ are hyperparameters. This continuous latent vector $\boldsymbol{x}$ is then quantized $f(\cdot)$ into one of the discrete latent codes $\boldsymbol{c} \in \mathcal{C}$ from the (fixed) codebook $\mathcal{C}$, using finite scalar quantization (FSQ, Mentzer et al. (2024)). As we have a discrete latent space, we formulate the transition dynamics to model the distribution over the next latent state $\boldsymbol{c}'$ given the previous latent state $\boldsymbol{c}$ and action $\boldsymbol{a}$. That is, we model stochastic transition dynamics in the latent space. We denote the probability of the next latent state $\boldsymbol{c}'$ taking the value of the $i^{\text{th}}$ code $\boldsymbol{c}^{(i)}$ as $p_i = p_\phi(\boldsymbol{c}' = \boldsymbol{c}^{(i)} \mid \boldsymbol{c}, \boldsymbol{a})$. This results in the next latent state following a categorical distribution $\boldsymbol{c}' \sim \text{Categorical}\left(p_1, \ldots, p_{|\mathcal{C}|}\right)$. We use a standard classification setup, where we use an MLP to predict the logits $\boldsymbol{l} = \{l_1, \ldots, l_{|\mathcal{C}|}\} = d_\phi(\boldsymbol{c}, \boldsymbol{a}) \in \mathbb{R}^{|\mathcal{C}|}$. Note that logits are the raw outputs from the final layer of the neural network (NN), which represent the unnormalized probabilities of the next latent state $\boldsymbol{c}'$ taking the value of each discrete code in the codebook $\mathcal{C}$. The logit for the $i^{th}$ code is given by $l_i = [d_\phi(\boldsymbol{c}, \boldsymbol{a})]_i \in \mathbb{R}$. We then apply softmax to obtain the probabilities $\{p_i\}_{i=1}^{|\mathcal{C}|}$ of the next latent state taking each discrete code in the codebook $\mathcal{C}$, *i.e.*, $p_i = \text{softmax}_i(\boldsymbol{l})$. DC-MPC utilizes the discrete codes $\boldsymbol{c}$ as its latent state for future predictions and decision-making.

**Quantized latent space** DCWM uses a discretized latent space where world states are encoded as discrete codes from a codebook $\mathcal{C}$. We use latent quantization to enforce data compression and encourage organization (Hsu et al., 2023). However, we implement this using finite scalar quantization (FSQ, Mentzer et al. (2024)) instead of dictionary learning (van den Oord et al., 2017). As a result, our codebook is fixed and we obviate two codebook learning loss terms, which stabilizes early training. In this section, we will give an overview of our discretization method which utilizes codebooks. First, let us assume the output of the encoder is a tensor[1] $\mathbf{x} \in \mathbb{R}^{b \times d}$, with $d$ dimensions and $b$ as the number of channels.

Each latent dimension is quantized into a codebook $\mathcal{C}$. That is, we have $d$ independent codebooks, one for each latent dimension. Our first step is to define the size of the codebook for each dimension, *i.e.* to define the ordered set of quantization levels $\mathcal{L} = \{L_1, L_2, \cdots, L_b\}$. Each quantization level $L_i$ corresponds to the $i$-th channel, *e.g.* $L_1$ defines the number of discrete values in the first channel, $L_2$ for the second and so on. In short, a quantization level of *e.g.* $L_i = 11$ would mean that we discretize each dimension in the $i$-th channel into 11 distinct values/symbols. We use integers as symbols, which would mean that the code for dimension $d$ in channel $i$ would be a symbol from the set $\{-5, -4, \cdots, 0, \cdots, 4, 5\}$. In practice, for fast conversion from continuous values to codes we use a similar discretization scheme as FSQ and apply the function

$$f : \boldsymbol{x}, \mathcal{L}, i \rightarrow \text{round}\left(\left\lfloor\frac{L_i}{2}\right\rfloor \cdot \tanh(\boldsymbol{x}_{i,:})\right), \quad (7)$$

to each channel, taking the output $\boldsymbol{x}$ of the encoder

Figure 2: **Illustration of Codebook** ($\mathcal{C}$) FSQ's codebook is a $b$-dimensional hypercube (left). This figure illustrates a $b$=3-dimensional codebook, where each axis of the 3-dimensional hypercube (left) corresponds to one dimension of the codebook (right). The $i^{\text{th}}$ dimension of the hypercube is discretized into $L_i$ values, *e.g.*, the $x$ and $y$-axis are discretized into $L_0 = L_1 = 5$ and the $z$-axis into $L_3 = 4$. Code symbols (here integers) are normalized to the range $[-1, 1]$.

and the channel quantization level $L_i$. This approach results in a codebook with $|\mathcal{C}| = \prod_{i=1}^{b} L_i$ unique codes for each dimension $d$, each code being made of $b$ symbols, *i.e.* a $b$-dimensional vector.

Intuitively, this results in a Voronoi partition of the $b$-dimensional space in each dimension $d$, where any point in space is assigned to one of the equidistantly placed centroids via Eq. (7). See Fig. 2 for a visualization. In effect, this leads to an efficient and fast discretization of the latent embedding space.

In practice, Eq. (7) is not differentiable. To solve this for using standard deep learning libraries, we use the straight-through gradient estimation (STE) approach with $\text{round\_ste}(\boldsymbol{x}) : x \rightarrow x + \text{sg}(\text{round}(\boldsymbol{x}) - \boldsymbol{x})$, where the function $\text{sg}(\cdot)$ stops the gradient flow. Furthermore, we normalize codes to be in the range $[-1, 1]$ after the discretization step as improved performance was reported by Mentzer et al. (2024). The hyperparameters of this approach are the number of channels $b$ and the number of code symbols per channel $L_i$, *i.e.* quantization levels. In our experiments, we found the quantization levels $\mathcal{L} = \{5, 3\}$ (*i.e.* $b = 2$ channels) to be sufficient.

**World model training** We train our world model components $e_\theta, d_\phi, R_\xi$ jointly using backpropagation through time (BPTT) with the following objective

$$\mathcal{L}(\theta, \phi, \xi; \mathcal{D}) = \mathbb{E}_{(\boldsymbol{o}, \boldsymbol{a}, \boldsymbol{o}', r)_{0:H} \sim \mathcal{D}} \left[ \sum_{h=0}^{H} \gamma^h \Big( \text{CE}(\underbrace{p_\phi(\hat{\boldsymbol{c}}_{h+1} \mid \hat{\boldsymbol{c}}_h, \boldsymbol{a}_h), \boldsymbol{c}_{h+1}}_{\text{Latent-state consistency}}) + \underbrace{\|R_\xi(\hat{\boldsymbol{c}}_h, \boldsymbol{a}_h) - r_h\|_2^2}_{\text{Reward prediction}} \Big) \right]$$

$$\text{with } \underbrace{\hat{\boldsymbol{c}}_0 = f(e_\theta(\boldsymbol{o}_0))}_{\text{First latent state}} \quad \underbrace{\hat{\boldsymbol{c}}_{h+1} \sim p_\phi(\hat{\boldsymbol{c}}_{h+1} \mid \hat{\boldsymbol{c}}_h, \boldsymbol{a}_h)}_{\text{Stochastic dynamics}} \quad \underbrace{\boldsymbol{c}_h = \text{sg}(f(e_\theta(\boldsymbol{o}_h)))}_{\text{Target latent code}}, \quad (8)$$

where $H$ denotes the multi-step prediction horizon and $\gamma$ is the discount factor. The first predicted latent code $\hat{\boldsymbol{c}}_0$ is obtained by passing the observation $\boldsymbol{o}_0$ through the encoder and then quantizing the output. At subsequent time steps, the dynamics model predicts the probability mass function over the next latent code $p_\phi(\hat{\boldsymbol{c}}_{h+1} \mid \hat{\boldsymbol{c}}_h, \boldsymbol{a}_h)$. Given this probabilistic dynamics model, we must consider how

---

[1]For simplicity, we omit here the batch dimension.

to make $H$-step predictions in the latent space. In practice, we propagate uncertainty by sampling and we use the straight-through (ST) Gumbel-softmax trick (Jang et al., 2017; Maddison et al., 2017) so that gradients backpropagate through our samples to the encoder. Note that gradients must flow back to the encoder at the first time step when it was used to obtain the first latent code $\hat{c}_0$, as the target codes $c$ are obtained by passing the next observation $o'$ through the encoder and using the stop gradient operator sg. We then train our dynamics "classifier" using the cross-entropy (CE) loss. Finally, we note that our reward model $R_\xi$ is trained jointly with the encoder $e_\theta$ and dynamics model $p_\phi$ to ensure that the world model can accurately predict rewards in the latent space.

**Policy and value learning**  We learn the policy $\pi_\eta(c)$ and action-value functions $q_\psi(c, a)$ in the latent space using the actor-critic RL method TD3 (Fujimoto et al., 2018). However, we follow Yarats et al. (2021b); Zhao et al. (2023) and augment the loss with $N$-step returns. The main difference to TD3 is that instead of using the original observations $o$, we map them through the encoder $c = f(e_\theta(o))$ and learn the actor/critic in the discrete latent space $c$. We also reduce bias in the TD target by following REDQ (Chen et al., 2021) and learning an ensemble of $N_q = 5$ critics, as was done in TD-MPC2. When calculating the TD target we randomly subsample two of the critics and use the minimum of these two. Let us denote the indices of the two randomly subsampled critics as $\mathcal{M}$. The critic is then updated by minimizing the following objective:

$$\mathcal{L}_q(\psi; \mathcal{D}) = \mathbb{E}_{(o,a,o',r)_{n=1}^N \sim \mathcal{D}} \left[ \frac{1}{N_q} \sum_{k=1}^{N_q} (q_{\psi_k}(\underbrace{f(e_\theta(o_t))}_{c_t}, a_t) - y)^2 \right], \tag{9}$$

$$y = \sum_{n=0}^{N-1} \gamma^n r_{t+n} + \gamma^N \min_{k \in \mathcal{M}} q_{\bar{\psi}_k}(\underbrace{f(e_\theta(o_{t+N}))}_{c_{t+N}}, a_{t+N}), \quad \text{with } a_{t+n} = \pi_{\bar{\eta}}(c_{t+n}) + \epsilon_{t+n},$$

where we use policy smoothing by adding clipped Gaussian noise $\epsilon_{t+n} \sim \text{clip}\left(\mathcal{N}(0, \sigma^2), -c, c\right)$ to the action $a_{t+n} = \pi_{\bar{\eta}}(c_{t+n}) + \epsilon_{t+n}$. We then use the target action-value functions $q_{\bar{\psi}}$ and the target policy $\pi_{\bar{\eta}}$ to calculate the TD target $y$. Note that the target networks use an exponential moving average, *i.e.* $[\bar{\psi}, \bar{\eta}] \leftarrow (1 - \tau)[\bar{\psi}, \bar{\eta}] + [\psi, \eta]$. We follow REDQ and learn the actor by minimizing

$$\mathcal{L}_\pi(\eta; \mathcal{D}) = -\mathbb{E}_{o_t \sim \mathcal{D}} \left[ \frac{1}{|\mathcal{M}|} \sum_{\psi_k \in \mathcal{M}} q_{\psi_k}(\underbrace{f(e_\theta(o_t))}_{c_t}, \pi_\eta(\underbrace{f(e_\theta(o_t))}_{c_t})) \right]. \tag{10}$$

That is, we train the actor to maximize the average action value over two subsampled critics.

**Summary**  Whilst this world model shares some similarities with TD-MPC2, there are some important distinctions. First, the latent space is represented as a discrete codebook which enables DC-MPC to train the dynamics model using the cross-entropy loss. Importantly, the cross-entropy loss considers a (potentially multimodal) distribution over the predicted latent codes during both training and inference. In contrast, TD-MPC2 considers deterministic dynamics and uses mean squared error regression. Interestingly, our experiments suggest that our stochastic dynamics model offers benefits in deterministic environments. Second, DC-MPC does not use value prediction when training the encoder. Instead, we follow the insight from Zhao et al. (2023) that value prediction is not necessary for obtaining a good latent representation and instead, train the action-value function separately.

Importantly, our discrete latent space is parameterized as a set of discrete codes from a codebook. It is worth highlighting that our codebook encoding preserves ordinal relationships between observations. This contrasts with one-hot encodings which were used by DreamerV2 (Hafner et al., 2022). See Sec. 3 for a comparison of the different discrete encodings. We hypothesize that this will offer significant improvements when representing continuous state vectors in a discrete space.

## 4.2 DECISION-TIME PLANNING

DC-MPC follows TD-MPC2 and leverages the world model for decision-time planning. It uses MPC to obtain a closed-loop controller and uses (modified) MPPI (Williams et al., 2015) as the underlying trajectory optimization algorithm (Betts, 1998; Scannell et al., 2021). MPPI is a sampling-based trajectory optimization method which does not require gradients. See Alg. 2 for full details. At each environment step, we estimate the parameters $\mu_{0:H}^*, \sigma_{0:H}^*$ of a diagonal multivariate Gaussian over a

$H$-step action sequence that maximizes the following objective

$$\boldsymbol{\mu}_{0:H}^*, \boldsymbol{\sigma}_{0:H}^* = \underset{\boldsymbol{\mu}_{0:H}, \boldsymbol{\sigma}_{0:H}}{\arg\max} \mathbb{E}_{\boldsymbol{a}_{0:H} \sim \mathcal{N}(\boldsymbol{\mu}_{0:H}, \text{diag}(\boldsymbol{\sigma}_{0:H}^2))} [J(\boldsymbol{a}_{0:H}, \boldsymbol{o})] \tag{11a}$$

$$J(\boldsymbol{a}_{0:H}, \boldsymbol{o}) = \sum_{h=0}^{H-1} \gamma^h R_\xi(\hat{\boldsymbol{c}}_h, \boldsymbol{a}_h) + \gamma^H \frac{1}{|\mathcal{M}|} \sum_{\psi_k \in \mathcal{M}} q_{\psi_k}(\hat{\boldsymbol{c}}_H, \boldsymbol{a}_H) \tag{11b}$$

$$\text{s.t.} \quad \hat{\boldsymbol{c}}_0 = f(e_\theta(\boldsymbol{o})) \quad \text{and} \quad \hat{\boldsymbol{c}}_{h+1} = \sum_{i=1}^{|\mathcal{C}|} \Pr(\hat{\boldsymbol{c}}_{h+1} = \boldsymbol{c}^{(i)} \,|\, \hat{\boldsymbol{c}}_h, \boldsymbol{a}_h) \boldsymbol{c}^{(i)}, \tag{11c}$$

where $H$ is the planning horizon and $\gamma$ is a discount factor. MPPI solves Eq. (11) in an iterative manner. It starts by sampling candidate action sequences and evaluating them using the objective $J(\boldsymbol{a}_{0:H}, \boldsymbol{o})$. It then refits the sampling distribution's parameters $\boldsymbol{\mu}_{0:H}, \boldsymbol{\sigma}_{0:H}^2$ based on a weighted average. After several iterations, we select an action trajectory and apply its first action $\boldsymbol{a}_0^{(i^*)}$ in the environment. Note that during training we promote exploration by adding Gaussian noise. Importantly, Eq. (11) uses the action-value function $\boldsymbol{q}_\psi(\boldsymbol{c}, \boldsymbol{a})$ to bootstrap the planning horizon such that it estimates the full RL objective. DC-MPC follows TD-MPC2 and warm starts the planning process with $N_\pi$ action sequences originating from the prior policy $\pi_\eta$ and we warm start by initializing $\boldsymbol{\mu}_{0:H}, \boldsymbol{\sigma}_{0:H}^2$ as the solution to the previous time step shifted by one. See App. A and Alg. 2 for further details.

Note that at planning time, we do not sample from the transition dynamics $p(\boldsymbol{c}_{h+1} \,|\, \boldsymbol{c}_h, \boldsymbol{a}_h)$ because this introduces unwanted stochasticity. Instead, we take the expected code, which is a weighted sum over the codes in the codebook. Whilst the expected value of a discrete variable does not necessarily take a valid discrete value, we find it effective in our setting. This is likely because our discrete codes have an ordering such that expected values simply interpolate between the codes in the codebook.

## 5    EXPERIMENTS

In this section, we experimentally evaluate DC-MPC in a variety of continuous control tasks from the DeepMind Control Suite (DMControl) (Tassa et al., 2018), Meta-World (Yu et al., 2019) and MyoSuite (Vittorio et al., 2022) against a number of baselines and ablations. Our experiments seek to answer the following research questions:

RQ1  Does DC-MPC's discrete latent space offer benefits over a continuous latent space?
RQ2  What is important for learning a latent space: *(i)* classification loss, *(ii)* discrete codebook, *(iii)* stochastic dynamics or *(iv)* multimodal dynamics?
RQ3  Does DC-MPC's codebook offer benefits for dynamics/value/policy learning over alternative discrete encodings such as *(i)* one-hot encoding (similar to DreamerV2) and *(ii)* label encoding?
RQ4  How does DC-MPC compare to state-of-the-art model-based RL algorithms leveraging latent state embeddings, especially in the hard DMControl and Meta-World tasks?

**Experimental Setup**   We compared DC-MPC against two state-of-the-art model-based RL baselines, namely DreamerV3 (Hafner et al., 2023) which utilizes a discrete one-hot encoding as its latent state and TD-MPC2 (Hansen et al., 2023) using a continuous latent space. We also compare against soft actor-critic (SAC) (Haarnoja et al., 2018), a model-free RL baseline, and the original TD-MPC (Hansen et al., 2022). Our proposed approach utilized a latent space with $d = 512$ dimensions and $b = 2$ channels, with 15 code symbols per dimension by using FSQ levels $\mathcal{L} = \{L_1 = 5, L_2 = 3\}$.

### 5.1    COMPARISON OF DIFFERENT LATENT SPACES

We first evaluate how different latent dynamics formulations affect the performance. We seek to answer the following: *(i)* do discrete latent spaces offer benefits over continuous latent spaces? *(ii)* does training with classification (cross-entropy) offer benefits over mean squared error regression? and *(iii)* does modeling stochastic (and potentially multimodal) transition dynamics offer benefits?

In our experiments, we consider both continuous and discrete latent spaces to investigate the impact of discretizing the latent space of the world model. In Figs. 3 and 9, the experiments with discrete latent spaces are labelled with "Discrete" (red, green, and purple) whilst continuous latent spaces are labelled "Continuous" (orange). We also evaluate DC-MPC using the simplical normalization used

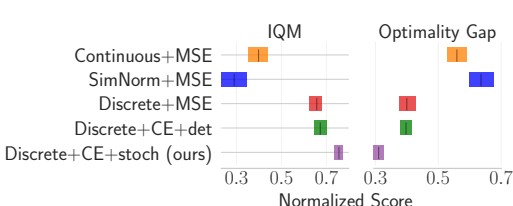 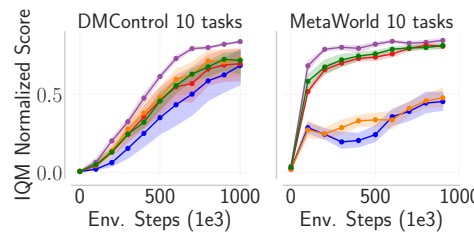

(a) **Aggregate statistics at 500k environment steps**    (b) **Training curves**

Figure 3: **Latent space ablation** Evaluation of *(i)* discrete (Discrete) vs continuous (Continuous) latent spaces, *(ii)* using cross-entropy (CE) vs mean squared error (MSE) for the latent-state consistency loss, and *(iii)* formulating a deterministic (det) vs stochastic (stoch) dynamics model. Discretizing the latent space (red) improves sample efficiency over the continuous latent space (orange) and formulating stochastic dynamics and training with cross-entropy (purple) improves performance further.

in TD-MPC2 – which bounds the latent space – labelled "SimNorm" (blue) . Experiments labelled with "MSE" were trained with mean squared error regression whilst those labelled "CE" were trained with the cross-entropy classification loss. The experiment labelled "Discrete+CE+det" used FSQ to get a discrete latent space and trained with the cross-entropy loss, where the logits were obtained as the MSE between the dynamics prediction and each code in the codebook. This experiment enabled us to test if DC-MPC's performance boost resulted from training with the cross-entropy loss or from making the dynamics stochastic. In Fig. 9, experiments labelled with "log-lik." were trained by maximizing the log-likelihood, *i.e.* cross-entropy for "FSQ-log-lik." (purple), Gaussian log prob. for "Gaussian+log-lik." (blue), and Gaussian mixture log prob. for "GMM+log-lik." (green).

**Discrete vs continuous latent spaces**   The experiments using discrete latent spaces (red and purple) significantly outperform the ones with continuous latent spaces in terms of sample efficiency. This suggests that our discrete codebook encoding offers significant benefits over continuous latent spaces.

**Classification vs regression**   Interestingly, training a deterministic discrete latent space using MSE regression (red) does not perform as well as training a stochastic discrete latent space using classification (purple). However, our experiment with the deterministic discrete latent space using classification (green) confirms that the benefit arises from the stochasticity of our latent space. This suggests that using straight-through Gumbel-softmax sampling (Jang et al., 2017) when making multi-step dynamics predictions during training boosts performance. Our results extending TD-MPC2 to use DC-MPC's discrete stochastic latent space in Fig. 6 support this conclusion.

**Deterministic vs stochastic**   Given that modeling a stochastic latent space and training with maximum log-likelihood is beneficial for discrete latent spaces, we now test if this holds in continuous latent spaces. To this end, we formulate two stochastic continuous latent spaces and compare them in Fig. 9. The first models a unimodal Gaussian distribution (blue) whilst the second models a multimodal Gaussian mixture model (GMM) (green). Interestingly, these stochastic transition models sometimes increase sample efficiency on DMControl tasks when compared to their deterministic counterparts (orange). However, they drastically underperform on Meta-World tasks.

Our method (purple) has a discrete latent space, is trained by maximum log-likelihood (*i.e.* cross-entropy), and models a (potentially multimodal) distribution over the latent transition dynamics during training. These factors, combined with using ST Gumbel-softmax sampling, offer improved sample efficiency over continuous latent spaces.

### 5.2    IMPACT OF LATENT SPACE ENCODING

Our world model consists of NNs for the dynamics $p_\phi(\mathbf{c}' \mid \mathbf{c}, \mathbf{a})$, reward $R_\xi(\mathbf{c}, \mathbf{a})$, critic $Q_\psi(\mathbf{c}, \mathbf{a})$, and prior policy $\pi_\eta(\mathbf{c})$, which all make predictions given the discrete codebook encoding $\mathbf{c} = \mathbf{e}_{\text{codes}}$. In Fig. 4, we evaluate what happens when we replace the codebook encoding $\mathbf{c}$ with *(i)* label encoding $e_{\text{label}} = i \in \{1, \ldots, |\mathcal{C}|\}$ and *(ii)* one-hot encoding $\mathbf{e}_{\text{one-hot}} = \mathbf{v} \in \{0, 1\}^{|\mathcal{C}|}$ given $\sum_{i=1}^{|\mathcal{C}|} v_i = 1$. In these experiments, we did not modify the dynamics $p_\phi(\mathbf{c}' \mid \mathbf{c}, \mathbf{a})$, that is, the dynamics continued to make predictions using the codebook encoding $\mathbf{c}$ and did not use the one-hot or label encodings. This is because when we replaced the codebook encoding with either one-hot or label encodings, this led to the training curves (environment step vs episode return) flat-lining and unable to solve the

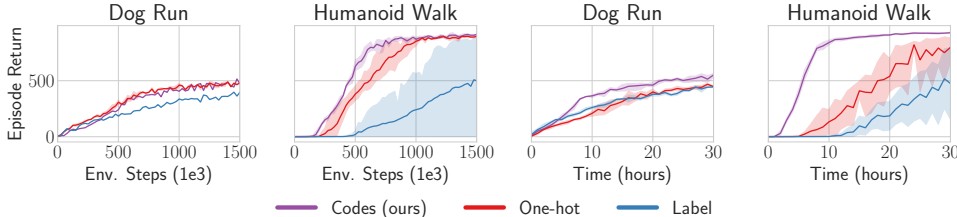

Figure 4: **Discrete encodings ablation** DC-MPC with its discrete codebook encoding (purple) outperforms using DC-MPC with one-hot encoding (red) and label encoding (blue), in terms of both sample efficiency (left) and computational efficiency (right). Dynamics model used codes $p_\phi(\mathbf{c}' \mid \mathbf{c}, \mathbf{a})$ whilst reward $R_\xi(\mathbf{e}, \mathbf{a})$, critic $Q_\psi(\mathbf{e}, \mathbf{a})$ and prior policy $\pi_\eta(\mathbf{e})$ used the respective encoding $\mathbf{e}$.

task. This suggests that our codebook encoding is needed in our self-supervised world model setup. Nevertheless, we evaluated the performance when changing the encoding for the other components.

We evaluated the following experiment configurations: **Codes (purple)**: All components used codes: dynamics $p_\phi(\mathbf{c}' \mid \mathbf{c}, \mathbf{a})$, reward $R_\xi(\mathbf{c}, \mathbf{a})$, critic $Q_\psi(\mathbf{c}, \mathbf{a})$ and prior policy $\pi_\eta(\mathbf{c})$. **Label (blue)**: Dynamics model used codes $p_\phi(\mathbf{c}' \mid \mathbf{c}, \mathbf{a})$ whilst reward $R_\xi(\mathbf{e}_{\text{label}}, \mathbf{a})$, critic $Q_\psi(\mathbf{e}_{\text{label}}, \mathbf{a})$ and prior policy $\pi_\eta(\mathbf{e}_{\text{label}})$ used labels $\mathbf{e}_{\text{label}}$ obtained from the code's index $i$ in the codebook. **One-hot (red)**: Dynamics model used codes $p_\phi(\mathbf{c}' \mid \mathbf{c}, \mathbf{a})$ whilst reward $R_\xi(\mathbf{e}_{\text{one-hot}}, \mathbf{a})$, critic $Q_\psi(\mathbf{e}_{\text{one-hot}}, \mathbf{a})$ and prior policy $\pi_\eta(\mathbf{e}_{\text{one-hot}})$, used the one-hot $\mathbf{e}_{\text{one-hot}}$ representation of the label encoding.

The label encoding (blue) struggles to learn in the Humanoid Walk task and is often less sample efficient than the alternative encodings. This is likely because the label encoding is not expressive enough to model the multi-dimensional ordinal structure of our codebook. Let us provide intuition via a simple example. Our codebook has $b = 2$ channels, so two different codes may take the form $e_{\text{codes}}(A) = [0.5, -0.5]$ and $e_{\text{codes}}(B) = [0, 0.5]$. As a result, our codebook encoding can model ordinal structure in both of its channels, *i.e.*, $e_{\text{codes}}(A)_1 > e_{\text{codes}}(B)_1$ whilst $e_{\text{codes}}(A)_2 < e_{\text{codes}}(B)_2$. The corresponding label encoding would encode this as $e_{\text{label}}(A) = 1$ and $e_{\text{label}}(B) = 2$, which incorrectly implies that $B > A$. In short, the label encoding cannot model the multi-dimensional ordinal structure of the codebook $\mathcal{C}$. In contrast, the one-hot encoding (red) matches the codebook encoding in terms of sample efficiency in all tasks except Humanoid Walk. However, the one-hot encoding introduces an extremely large input dimension for the reward, value and policy networks, and this significantly slows down training. See Sec. 3 for further details on why this is the case.

### 5.3 PERFORMANCE OF DC-MPC

In Figs. 5, 14, 16 and 18, we compare the aggregate performance of DC-MPC against TD-MPC2, DreamerV3, TD-MPC, and SAC, in 30 DMControl, 45 Meta-World, and 5 MyoSuite tasks respectively, with 3 seeds per task. Some tasks in DMControl are particularly high-dimensional. For instance, the observation space of the Dog tasks is $\mathcal{O} \in \mathbb{R}^{223}$ and the action space is $\mathcal{A} \in \mathbb{R}^{38}$, and for Humanoid, the observation space is $\mathcal{O} \in \mathbb{R}^{67}$ and the action space $\mathcal{A} \in \mathbb{R}^{24}$. Fig. 13 shows that DC-MPC excels in the high dimensional Dog and Humanoid environments when compared to the baselines. We hypothesize that our discretized representations are particularly beneficial for simplifying learning the transition dynamics in high-dimensional spaces, making DC-MPC highly sample efficient in these tasks. Similarly, we find that DC-MPC outperforms DreamerV3 in simulated manipulation tasks in the Meta-World task suite (Figs. 5, 15 and 16). We also see that DC-MPC

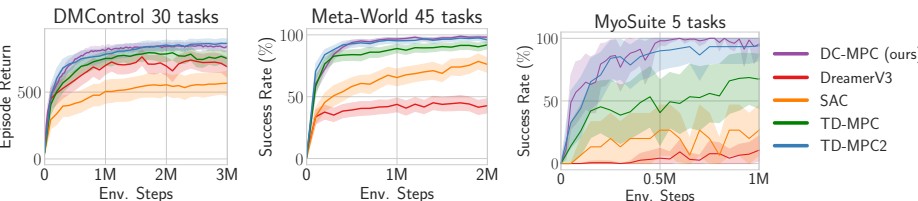

Figure 5: **Aggregate training curves in DMControl, Meta-World, & MyoSuite** DC-MPC generally matches TD-MPC2 whilst outperforming DreamerV3, SAC and TD-MPC across all tasks. We plot the mean (solid line) and the $95\%$ confidence intervals (shaded) across 3 seeds per task.

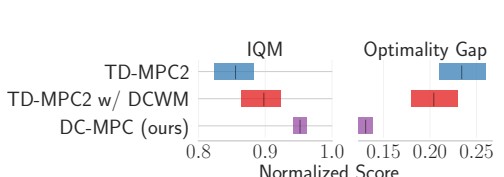

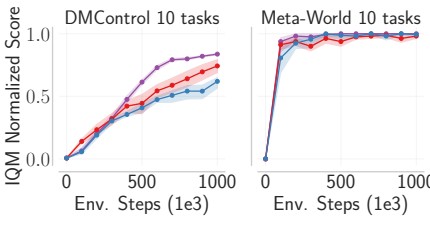

(a) **Aggregate statistics at 1M environment steps**    (b) **Training curves**

Figure 6: **TD-MPC2 with DCWM** Adding DC-MPC's discrete and stochastic latent space to TD-MPC2 improves performance. See Apps. B and B.10 for more details.

generally matches the performance of TD-MPC2. Comparing the results at a global level (Fig. 5), we can find that our proposed method performs well across all benchmarks.

It is important to note that TD-MPC2 has multiple algorithmic differences to DC-MPC which means that a straight-up comparison between them is not only affected by the latent space design. For example, it *(i)* uses soft-actor critic (SAC) to learn the prior policy (helping with exploration in sparse reward tasks), *(ii)* learns the value function jointly with the world model, and *(iii)* the reward and value functions are formulated using discrete regression in a $\log$-transformed space. In Fig. 6, we show that incorporating DCWM's stochastic and discrete codebook latent space into TD-MPC2 (red) offers improvements over vanilla TD-MPC2. See Apps. B and B.10 for more details on these experiments and App. B.9 where we tried the same experiments with DreamerV3. DreamerV3's performance is poor in the harder tasks so we do not see any benefit from using DCWM. However, we identify that its poor performance stems from using observation reconstruction.

**Further Experiments**  In Apps. B.1 and B.3, we evaluate DC-MPC's sensitivity to codebook size $|\mathcal{C}|$ and latent dimension $d$, respectively, in App. B.3, we show that stochastic continuous latent spaces do not appear to offer the same benefits as stochastic discrete latent spaces, in App. B.4, we ablate FSQ and show that it either matches or outperforms vector quantization (VQ) whilst being simpler, and in App. B.5, we show the benefit of using REDQ's ensemble of critics vs the standard double Q approach.

## 6   CONCLUSION

We have presented DC-MPC, a world model that learns a discrete and stochastic latent space using codebook encodings and a cross-entropy-based self-supervised loss for model-based RL. DC-MPC demonstrates strong performance in continuous control tasks, including Meta-World and the complex DMControl Humanoid and Dog tasks, where it exceeds or matches the performance of SOTA baselines. Our results indicate that using straight-through Gumbel-softmax sampling when making multi-step dynamics predictions is beneficial for world model learning, both in DC-MPC and our experiments where we modified TD-MPC2's latent space. In summary, we have demonstrated the benefit of a discrete latent space with codebook encodings over a standard continuous latent embedding or classical discrete spaces such as label and one-hot encodings. These findings open up a new interesting avenue for future research into discrete embeddings for world models.

**Limitations and Future Work**  As our goal was to evaluate latent space design, we did not prioritize making DC-MPC run with a single set of hyperparameters and we tuned the noise schedule and the $N$-step return for some tasks. In future work, it would be interesting to make DC-MPC robust to hyperparameters. For example, it would be interesting to model the epistemic uncertainty associated with the latent transition dynamics – arising from learning from limited data – and using it to equip DC-MPC with a more principled exploration mechanism like Chua et al. (2018); Scannell et al. (2024c); Daxberger et al. (2021) and remove the task-specific noise schedules. It would also be interesting to investigate if our results hold for different world model backbones (Deng et al., 2023; NVIDIA et al., 2025), such as Transformers (Vaswani et al., 2017; Robine et al., 2022; Zhang et al., 2023; Micheli et al., 2022; Bar et al., 2024) and diffusion models (Ho et al., 2020; Alonso et al., 2024). Finally, it would be interesting to investigate how well DC-MPC scales (Kaplan et al., 2020; Henighan et al., 2020; Hoffmann et al., 2022) and if it is an effective setup for generalist (*i.e.* multi-embodiment) world modeling (Reed et al., 2022; Zhao et al., 2025).

ACKNOWLEDGMENTS

Aidan Scannell and Kalle Kujanpää were supported by the Research Council of Finland from the Flagship program: Finnish Center for Artificial Intelligence (FCAI). Arno Solin and Yi Zhao acknowledge funding from the Research Council of Finland (grant ids 339730 and 357301, respectively) and Mohammadreza Nakhaei acknowledges funding from Business Finland (BIOND4.0 – Data Driven Control for Bioprocesses). Kevin Sebastian Luck is supported by the project *TeNet: Text-to-Network for Fast and Energy-Efficient Robot Control* with file number NGF.1609.241.015 of the research programme National Growth Fund AiNed XS Europe 24-2 which is financed by the Dutch Research Council (NWO). We acknowledge CSC – IT Center for Science, Finland, for awarding this project access to the LUMI supercomputer, owned by the EuroHPC Joint Undertaking, hosted by CSC (Finland) and the LUMI consortium through CSC. We acknowledge the computational resources provided by the Aalto Science-IT project.

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

APPENDICES

This appendix is organized as follows. In App. A we provide further details on our method. App. B provides further experimental results, including evaluating DC-MPC's sensitivity to the codebook size in App. B.1, its sensitivity to latent dimension in App. B.2, further details on the latent space ablation in App. B.3, a comparison of DC-MPC using VQ instead of FSQ in App. B.4, a comparison of DC-MPC's ensemble REDQ critic approach vs the standard double Q approach in App. B.5, full DeepMind control suite results in App. B.6, Meta-World results in App. B.7, MyoSuite results in App. B.8, evaluation of DreamerV3 using DC-MPC's latent space in App. B.9 and an evaluation of TD-MPC2 using DC-MPC's latent space in App. B.10. In App. C, we provide further implementation details, including default hyperparameters, hardware, *etc.* In App. D, we provide further details of the baselines and in App. E we detail the different DeepMind control, Meta-World and MyoSuite tasks used throughout the paper.

# A  METHOD DETAILS

Alg. 1 outlines DC-MPC's training procedure.

---

**Algorithm 1** DC-MPC's training

---

**Input:** Encoder $e_\theta$, dynamics $d_\phi$, reward $R_\xi$, critics $\{q_{\psi_i}\}_{i=1}^{N_q}$, policy $\pi_\eta$, learning rate $\alpha$, target network update rate $\tau$, episode length $T$, replay buffer $\mathcal{D} = \{\}$

**for** $1 : N_{\text{random episodes}}$ **do**
    $\mathcal{D} \leftarrow \mathcal{D} \cup \{\boldsymbol{o}_t, \boldsymbol{a}_t, \boldsymbol{o}_{t+1}, r_t\}_{t=0}^T$                 ▷ Collect data using random policy
**end for**
**for** $1 : N_{\text{episodes}}$ **do**
    $\mathcal{D} \leftarrow \mathcal{D} \cup \{\boldsymbol{o}_t, \boldsymbol{a}_t, \boldsymbol{o}_{t+1}, r_t\}_{t=0}^T$                   ▷ Collect data using DC-MPC
    **for** $i = 1$ **to** $T$ **do**
        $[\theta, \phi, \xi] \leftarrow [\theta, \phi, \xi] + \alpha \nabla \left( \mathcal{L}(\theta, \phi, \xi; \mathcal{D}) \right)$       ▷ Update world model, Eq. (8)
        $\psi \leftarrow \psi + \alpha \nabla \left( \mathcal{L}_q(\psi; \mathcal{D}) \right)$                     ▷ Update critic, Eq. (9)
        **if** $i \% 2 == 0$ **then**
            $\eta \leftarrow \eta + \alpha \nabla \left( \mathcal{L}_\pi(\eta; \mathcal{D}) \right)$     ▷ Update actor less frequently than critic, Eq. (10)
        **end if**
        $[\bar{\psi}, \bar{\eta}] \leftarrow (1 - \tau)[\bar{\psi}, \bar{\eta}] + \tau[\psi, \eta]$               ▷ Update target networks
    **end for**
**end for**

---

Alg. 2 outlines how we perform trajectory optimization using MPPI (Williams et al., 2015), closely following the formulation of MPPI by Hansen et al. (2022), with two key modifications. First, during each rollout, we use the expected next latent state, *i.e.* a weighted sum over the codes in the codebook. Note that this contrasts our world model training where we sample from the transition dynamics $p(\boldsymbol{c}_{h+1}|\boldsymbol{c}_h, \boldsymbol{c}_h)$. This approach reduces the variance in state transitions, which results in more stable trajectory evaluations. Second, we do not add noise sampled from the standard deviation $\boldsymbol{\sigma}_0^2$ returned from MPPI. Instead, we promote exploration by adding noise sampled from a separate noise schedule. This method, inspired by TD3 (Fujimoto et al., 2018), strikes a better balance between exploration and exploitation, leading to more stable training performance.

It is worth noting that MPPI resembles the CEM-based planner in Chua et al. (2018), however, instead of simply fitting a Gaussian to the top $K$ action samples at each iteration, MPPI uses weighted importance sampling, which weights **all** samples by their empirical return estimates. However, we follow Hansen et al. (2022) and use a hybrid approach, which selects the top $K$ action samples (like CEM) but then use weighted importance sampling (like MPPI). At each iteration, we calculate the mean and variance of the action trajectory as follows,

$$\boldsymbol{\mu}_{0:H} = \text{fit\_mean}\left( \left\{ \left( \boldsymbol{a}_{0:H}^{(i)}, \Phi^{(i)} \right) \right\}_{i=0}^K \right) = \sum_{i=1}^K \underbrace{\frac{\Omega^{(i)}}{\sum_{j=1}^K \Omega^{(j)}}}_{\text{importance weight}} \boldsymbol{a}_{0:H}^{(i)} \tag{12}$$

$$\boldsymbol{\sigma}_{0:H}^2 = \text{fit\_var}\left( \left\{ \left( \boldsymbol{a}_{0:H}^{(i)}, \Phi^{(i)} \right) \right\}_{i=0}^K \right) = \frac{\sum_{i=1}^K \Omega^{(i)} \left( \boldsymbol{a}_{0:H}^{(i)} - \boldsymbol{\mu}_{0:H} \right)^2}{\sum_{i=1}^K \Omega^{(i)}} \tag{13}$$

where $\Omega^{(i)}$ is the exponentiated normalized empirical return estimate given by $\Omega^{(i)} = \exp\left( \tau_{\text{MPPI}} \left( \Phi^{(i)} - \max\left( \{\Phi^{(0)}, \ldots, \Phi^{(N_p + N_\pi)}\} \right) \right) \right)$. Note that $\tau_{\text{MPPI}}$ is the (inverse) temperature parameter and $\Phi^{(i)}$ denotes the return estimate for the $i^{\text{th}}$ action trajectory $\boldsymbol{a}_{0:H}^{(i)}$. After $J$ (default 6) iterations, we sample one of the top $K$ action sequences $\{\boldsymbol{a}_{0:H}^{(i)}\}_{i \in \{i_1^*, \ldots, i_K^*\}}$ where each action sequence is weighted by its empirical return estimate $\{\Omega^{(i)}\}_{i \in \{i_1^*, \ldots, i_K^*\}}$. We then apply the first action $\boldsymbol{a}_0^{(i^*)}$ in the environment.

–appendices continue on next page–

---

**Algorithm 2** DC-MPC's inference (modified MPPI)

---

**Input:** current observation $\boldsymbol{o}$, planning horizon $H$, iterations $J$, population size $N_p$, prior population size $N_\pi$, number of elites $K$, exploration noise std $\sigma_{\text{noise}}$

$\boldsymbol{c}_0 \leftarrow e_\theta(\boldsymbol{o})$ ▷ Encode observation into discrete code

Initialize $\boldsymbol{\mu}_{0:H}^0, (\boldsymbol{\sigma}_{0:H}^2)^0$ with the solution from the last time step shifted by one.

**for** each iteration $j = 1, \ldots, J$ **do**

    Sample $N_p$ action trajectories of length $H$ from $\{a_h \sim \mathcal{N}(\boldsymbol{\mu}_h^{j-1}, (\boldsymbol{\sigma}_h^2)^{j-1})\}_{h=0}^H$ ▷ Sample action candidates

    Sample $N_\pi$ action trajectories of length $H$ using $\pi_\eta$ and $d_\phi$ ▷ Prior policy samples

    **for** all $N_p + N_\pi$ action sequences $\left\{\tau^{(i)} = \left(\boldsymbol{a}_0^{(i)}, \ldots, \boldsymbol{a}_H^{(i)}\right)\right\}_{i=1}^{N_p + N_\pi}$ **do** ▷ Trajectory evaluation

        $\Phi^{(i)} \leftarrow 0$

        **for** step $h = 0, \ldots, H-1$ **do**

            $\Phi^{(i)} \leftarrow \Phi^{(i)} + \gamma^h R_\xi(\hat{\boldsymbol{c}}_h, \boldsymbol{a}_h^{(i)})$ ▷ Compute immediate reward

            $\hat{\boldsymbol{c}}_{h+1} = \sum_{k=1}^{|\mathcal{C}|} \Pr(\hat{\boldsymbol{c}}_{h+1} = \boldsymbol{c}^{(k)} \mid \hat{\boldsymbol{c}}_h, \boldsymbol{a}_h^{(i)}) \boldsymbol{c}^{(k)}$ ▷ Compute next state

        **end for**

        $\Phi^{(i)} \leftarrow \Phi^{(i)} + \gamma^H \frac{1}{N_q} \sum_{k=1}^{N_q} q_{\psi_k}(\boldsymbol{c}_H, \boldsymbol{a}_H^{(i)})$ ▷ Bootstrap with ensemble of Q-functions

    **end for**

    $\Phi^{(i_1^*)}, \ldots, \Phi^{(i_K^*)} = \text{topk}(\{\Phi^{(0)}, \ldots, \Phi^{(N_p + N_\pi)}\})$ ▷ Get top-$K$ elite scores

    $\boldsymbol{\mu}_{0:H} \leftarrow \text{fit\_mean}\left(\left\{\left(\boldsymbol{a}_{0:H}^{(i)}, \Phi^{(i)}\right)\right\}_{i \in \{i_1^*, \ldots, i_K^*\}}\right)$ ▷ Update mean of action dist.

    $\boldsymbol{\sigma}_{0:H}^2 \leftarrow \text{fit\_var}\left(\left\{\left(\boldsymbol{a}_{0:H}^{(i)}, \Phi^{(i)}\right)\right\}_{i \in \{i_1^*, \ldots, i_K^*\}}\right)$ ▷ Update variance of action dist.

**end for**

$i^* \sim \text{Categorical}\left(\text{softmax}(\{\Phi^{(i_1^*)}, \ldots, \Phi^{(i_K^*)}\})\right)$ ▷ Sample action index according to scores

**return** $\boldsymbol{a}_0^{(i^*)} + \epsilon$    with    $\epsilon \sim \mathcal{N}(0, \sigma_{\text{noise}}^2)$ ▷ Final output with exploration noise

---

# B FURTHER RESULTS

In this section, we include further results and ablations.

**Aggregate metrics** In Figs. 14, 16 and 18, we compare the aggregate performance of DC-MPC against TD-MPC, TD-MPC2, DreamerV3, and SAC, in 30 DMControl tasks, 45 Meta-World tasks, and 5 MyoSuite tasks respectively, with 3 seeds per task. Following Agarwal et al. (2021), we report the median, interquartile mean (IQM), mean, and optimality gap at 1M environment steps, with error bars representing $95\%$ stratified bootstrap confidence intervals. For DMControl, we use min-max normalization as the maximum possible return in an episode is $1000$ whilst the minimum is $0$, *i.e.* Normalized Return $=$ Return$/(1000 - 0)$. For Meta-World, we report the success rate which does not require normalization as it is already between $0$ and $1$.

In Figs. 3 and 6 we report aggregate metrics over 10 DMControl and 10 Meta-World tasks. The tasks are as follows:

- **DMControl 10**: Acrobot Swingup, Dog Run, Dog Walk, Dog Stand, Dog Trot, Humanoid Stand, Humanoid Walk, Humanoid Run, Reacher Hard, Walker Walk.
- **Meta-World 10**: Button Press, Door Open, Drawer Close, Drawer Open, Peg Insert Side, Pick Place, Push, Reach, Window Open, Window Close.

–appendices continue on next page–

## B.1 SENSITIVITY TO CODEBOOK SIZE $|\mathcal{C}|$

In this section, we evaluate how the size of the codebook $|\mathcal{C}|$ influences training. We indirectly configure different codebook sizes via the FSQ levels $\mathcal{L} = \{L_1, \ldots, L_b\}$ hyperparameter. This is because the codebook size is given by $|\mathcal{C}| = \prod_{i=1}^{b} L_i$. The top row of Fig. 7 compares the training curves for different codebook sizes. The algorithm's performance is not particularly sensitive to the codebook size. A codebook that is too large can result in slower learning. The best codebook size varies between environments.

Given that a codebook has a particular size, we can gain insights into how quickly DC-MPC's encoder starts to activate all of the codebook. The connection between the codebook size and the activeness of the codebook is intuitive: the bottom row of Fig. 7 shows that the smaller the codebook, the larger the active proportion.

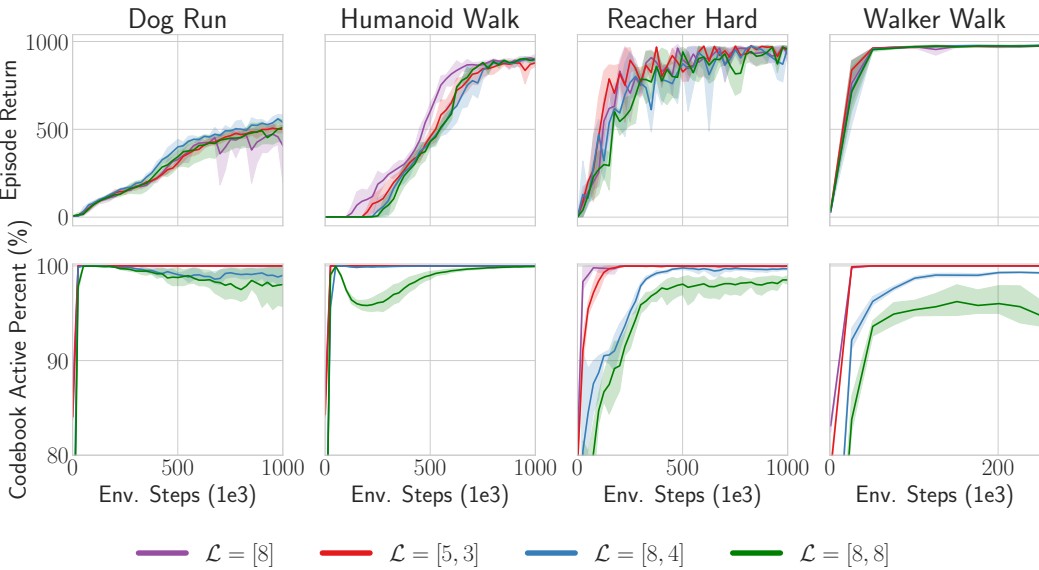

Figure 7: **Sensitivity to codebook size** We compare how the codebook size affects the performance of DC-MPC (top) and the percentage of the codebook that is active during training (bottom). In general, smaller codebooks become fully active faster than larger codebooks. We plot the mean and the 95% confidence intervals (shaded) across 3 random seeds for all environments.

–appendices continue on next page–

## B.2 SENSITIVITY TO LATENT DIMENSION $d$

This section investigates how the latent dimension $d$ affects the behavior and performance of DC-MPC in four different environments. In the top row of Fig. 8, we see that the performance of our algorithm is robust to the latent dimension $d$, although a latent dimension too small can result in inferior performance, especially in the more difficult environments. The bottom row of Fig. 8 demonstrates that DC-MPC learns to use the complete codebook irrespective of the latent dimension.

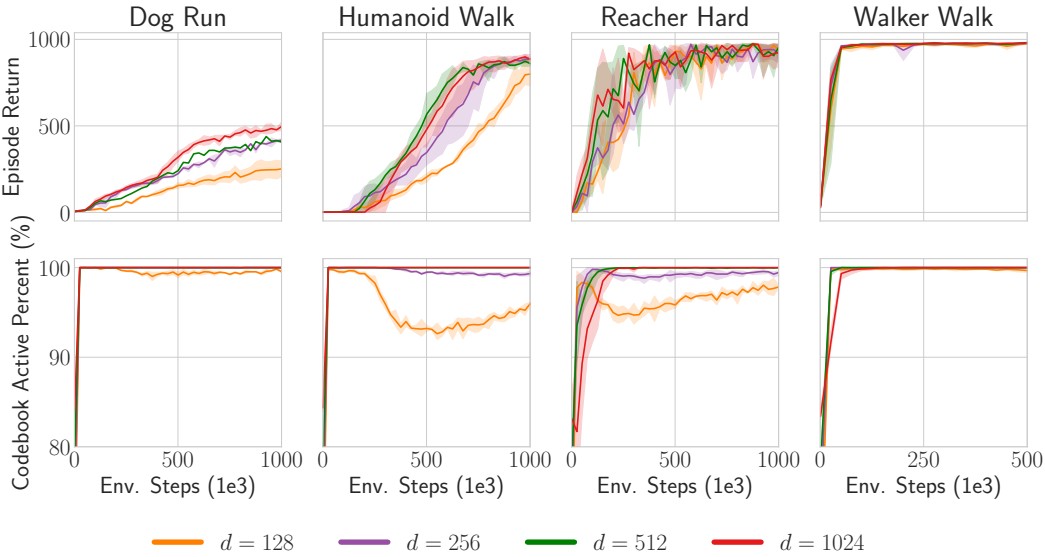

Figure 8: **Sensitivity to latent dim** $d$ We compare how the latent dimension $d$ affects the performance of DC-MPC (top) and the percentage of the codebook that is active during training (bottom). In general, our algorithm is robust to the latent dimension of the representation, although in more difficult environments, such as Humanoid Walk, a $d$ too small can harm the agent's performance. We plot the mean and the $95\%$ confidence intervals (shaded) across 3 random seeds for all environments.

## B.3 ABLATION OF LATENT SPACE

In this section, we provide further details on the comparison of different latent spaces experiments in Sec. 5.1. To validate our method, we test the importance of quantizing the latent space and training the world model with classification instead of regression. In Fig. 9, we compare DC-MPC to world models with different latent spaces formulations, which we now detail.

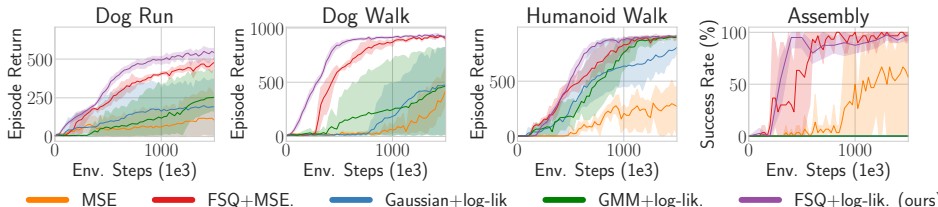

Figure 9: **Latent space comparison** Comparison of different latent space formulations. Continuous and deterministic latent space trained with MSE regression (orange), deterministic and discrete trained with MSE (red), continuous and unimodal Gaussian latent space trained with maximum log-likelihood (blue), continuous and multimodal GMM trained with maximum log-likelihood (green), and discrete trained with classification (purple). Discretizing the latent space with FSQ (red) improves sample efficiency and making the dynamics stochastic and training with classification (purple) improves performance further.

**MSE (orange)** First, we consider a continuous latent space with deterministic transition dynamics trained by minimizing the mean squared error between predicted next latent states and target next latent states.

**FSQ+MSE (red)** Next, we consider quantization of the latent space and training based on mean squared error regression. This experiment allows us to analyze the importance of quantization.

**Gaussian+log-lik. (blue)** To consider stochastic continuous dynamics, we configure the transition dynamics to model a Gaussian distribution over predictions of the next state. During training, we sample from the Gaussian distribution using the reparameterization trick. The world model is then trained to maximize the log-likelihood of the next latent state targets. This allows us to investigate if modeling stochastic transition dynamics offers benefits when using continuous latent spaces.

**GMM+log-lik. (green)** To consider continuous multimodal transitions, we consider a Gaussian mixture with three components. During training, we sample a Gaussian from the mixture with the ST Gumbel-softmax trick and then we sample from the selected Gaussian using the reparameterization trick. The world model is then trained to maximize the log-likelihood of next latent state targets.

–appendices continue on next page–

### B.4 ABLATION OF FSQ VS VECTOR QUANTIZATION (VQ)

To understand how the choice of using FSQ for discretization contributes to the performance of our algorithm, we tried replacing the FSQ layer with a standard Vector Quantization layer. We evaluated the methods in Walker Walk, Dog Run, Humanoid Walk, and Reacher Hard. We used standard hyperparameters, $\beta = 0.25$, and an EMA-updated codebook with a size of 256 and either 256 (dog) or 128 (other tasks) channels per dimension. We did not change other hyperparameters from DC-MPC. However, we found that to approach the performance of standard FSQ, VQ needs environment-dependent adjusting of the planning procedure. In Humanoid Walk, the performance of FSQ aligns closely with the VQ with a weighted sum over the codes in the codebook for planning (expected code) but significantly outperforms sampled VQ. Conversely, standard sampling is superior in Reacher Hard, which is unsurprising, as the discrete codes in VQ have not been ordered like in FSQ. The necessary environment-specific adjustments for VQ undermine its general applicability compared to FSQ.

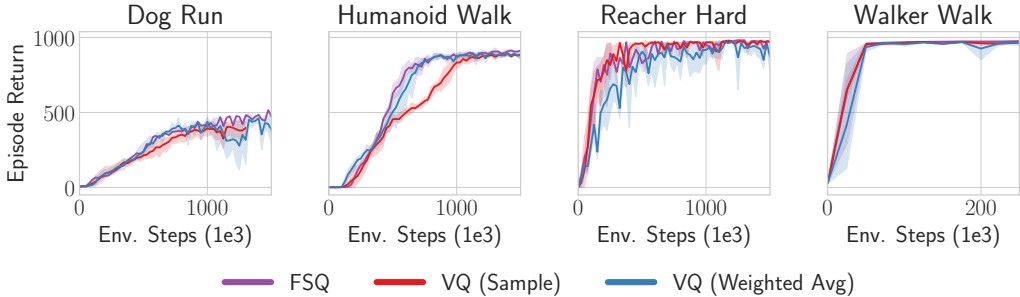

Figure 10: **Ablation of FSQ vs VQ** FSQ does not require the extra loss terms required by VQ and it generally performs equal to or better and VQ.

–appendices continue on next page–

## B.5 Ablation of REDQ Critic vs Standard Double Q Approach

In this section, we compare the ensemble of Q-functions approach, used by DC-MPC, REDQ (Chen et al., 2021) and TD-MPC2 (Hansen et al., 2023), to the standard double Q approach (Fujimoto et al., 2018). In Fig. 11, we evaluate how our default ensemble size of $N_q = 5$ (purple) compares with the standard double Q approach, which is obtained by setting the ensemble size to $N_q = 2$ (blue). Note that we always sample two critics so the $N_q = 2$ result reduces to the standard double Q approach. Fig. 11 shows that DC-MPC works fairly well with both approaches but the ensemble approach offers benefits in the harder Dog Run and Humanoid Walk tasks.

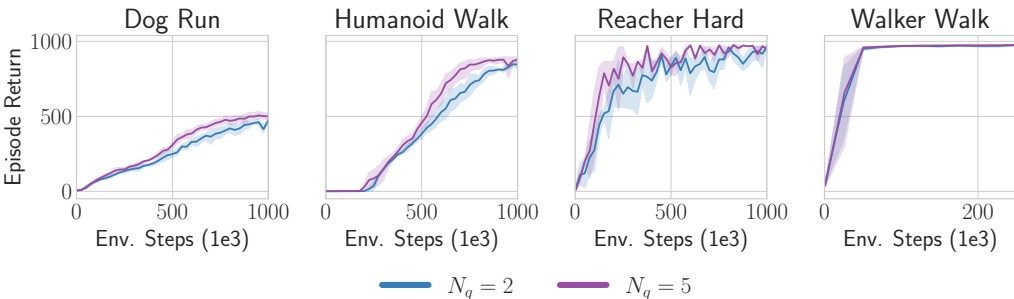

Figure 11: **Ablation of REDQ critic vs standard double Q** DC-MPC uses a Q ensemble, similar to REDQ, of size $N_q = 5$ (purple) and sub samples two critics when calculating the mean or minimum Q-value. We compare this approach to the standard double Q approach by setting $N_q = 2$ (blue) and we see that the ensemble approach offers a slight benefit in the harder Dog Run and Humanoid Walk.

–appendices continue on next page–

## B.6 DEEPMIND CONTROL RESULTS

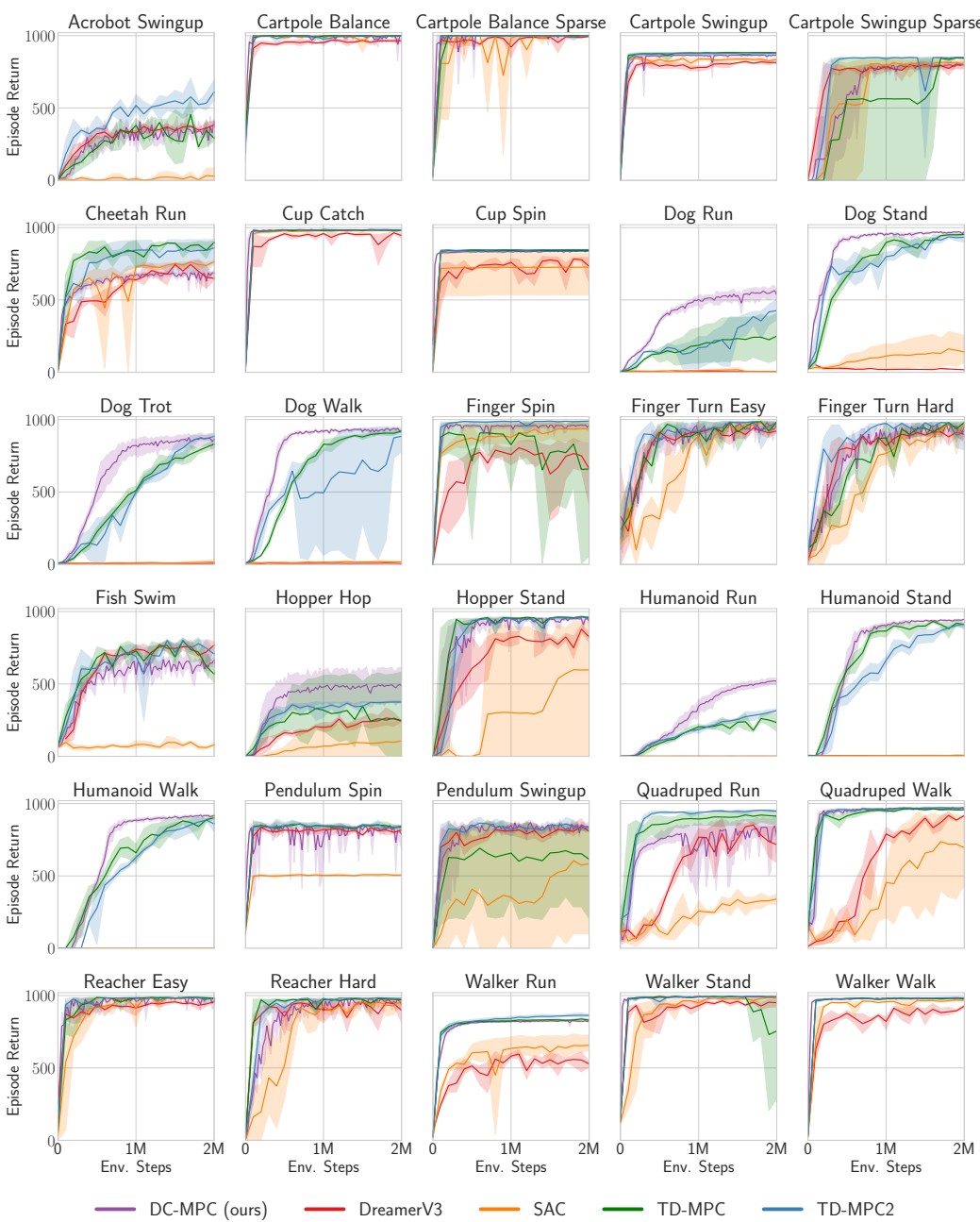

Figure 12: **DeepMind Control results.** DC-MPC performs well across a variety of DMC tasks. We plot the mean (solid line) and the $95\%$ confidence intervals (shaded) across 5 seeds (DC-MPC) or 3 seeds (TD-MPC2/TD-MPC/DreamerV3/SAC), where each seed averages over 10 evaluation episodes.

–appendices continue on next page–

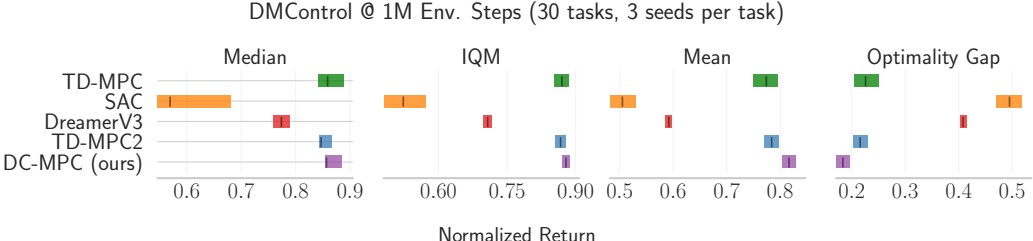

Figure 13: **High-dimensional locomotion** DC-MPC (purple) significantly outperforms TD-MPC2 (blue) and DreamerV3 (red) in the complex, high-dimensional locomotion tasks from DMControl.

Figure 14: **DMControl aggregate results** DC-MPC generally outperforms TD-MPC2 and DreamerV3 in DMControl tasks. This is due to DC-MPC's strong performance in the hard Dog and Humanoid tasks. Error bars represent 95% stratified bootstrap confidence intervals.

–appendices continue on next page–

## B.7 META-WORLD MANIPULATION RESULTS

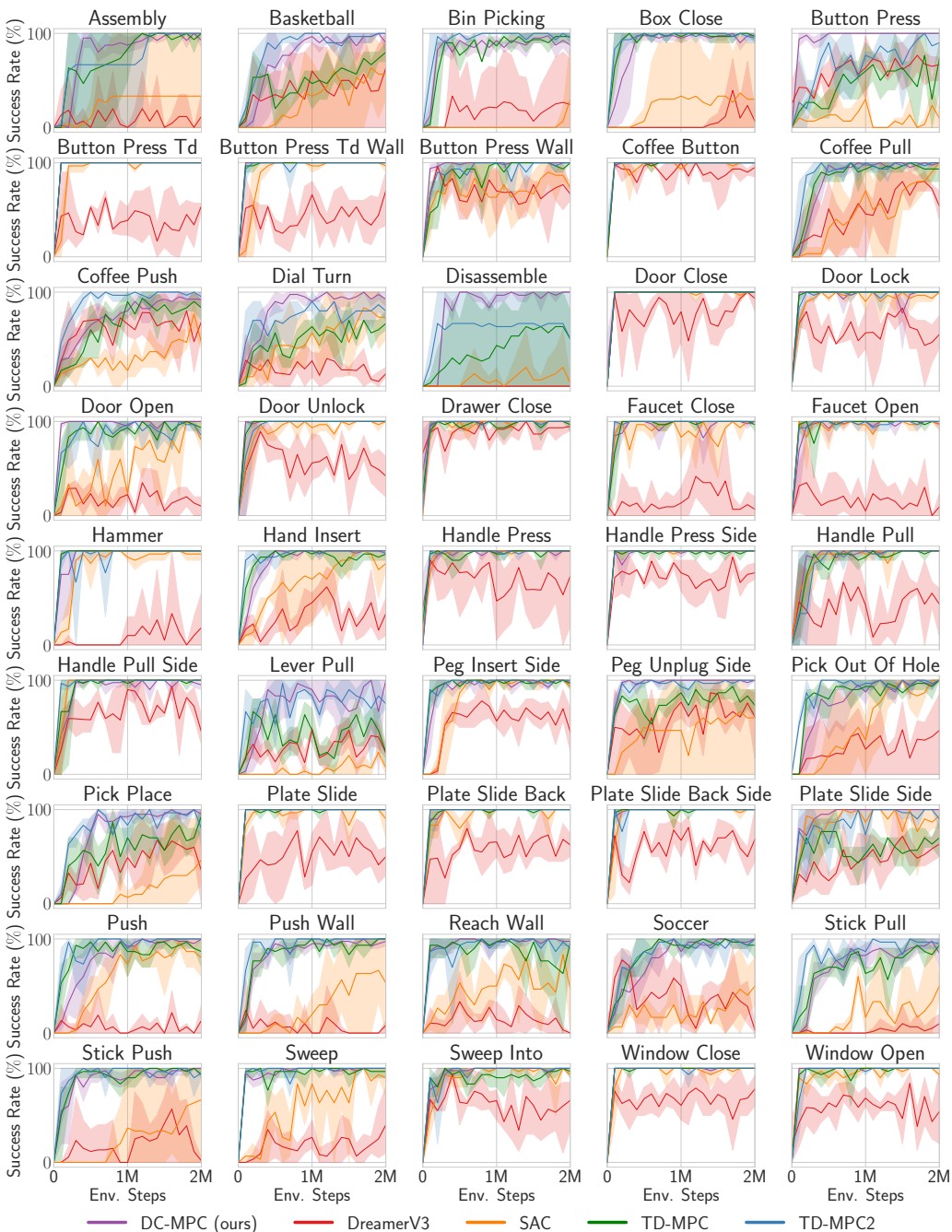

Figure 15: **Meta-World manipulation results** DC-MPC performs well across Meta-World tasks. We plot the mean (solid line) and the 95% confidence intervals (shaded) across 3 seeds, where each seed averages over 10 evaluation episodes.

–appendices continue on next page–

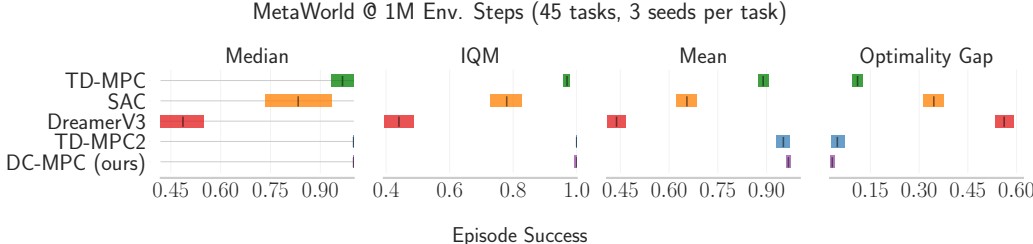

Figure 16: **Meta-World results** DC-MPC performs well in Meta-World, generally matching TD-MPC2, whilst significantly outperforming DreamerV3 and SAC. Error bars represent 95% stratified bootstrap confidence intervals.

–appendices continue on next page–

B.8 MYOSUITE MUSCULOSKELETAL RESULTS

In this section, we evaluate DC-MPC in five musculoskeletal tasks from MyoSuite.

In these experiments, we followed Hafner et al. (2023); Hansen et al. (2023) and scaled the rewards using symlog($\cdot$),

$$\text{symlog}(x) = \text{sign}(x)\ln(|x| + 1). \quad (14)$$

This compresses large and small rewards whilst preserving the input sign as it is a symmetric function. Note that we simply transform the rewards with symlog and learn both the reward function and $Q$-functions using these transformed rewards. We use $N = 1$-step returns in Hand Key Turn, Hand Obj Hold and Hand Pen Twirl and we use $N = 5$-step returns in Hand Pose and Hand Reach. In Hand Pose we also had to adjust the temperature from $0.5$ to $0.2$. In future work, it would be interesting to investigate if using $\lambda$-returns – which uses a weighted-sum of $N$-step returns – can make DC-MPC robust to the $N$-step hyperparameter. Further to this, it would be interesting to explore methods for dynamically tuning the MPPI (inverse) temperature $\tau_{\text{MPPI}}$.

In Fig. 17 we show the training curves for the individual tasks. Fig. 18 then reports aggregate metrics at 1M environment steps over three random seeds in the five tasks. On average, DC-MPC performs well, generally matching TD-MPC2 at 1M environment steps and outperforming the other baselines.

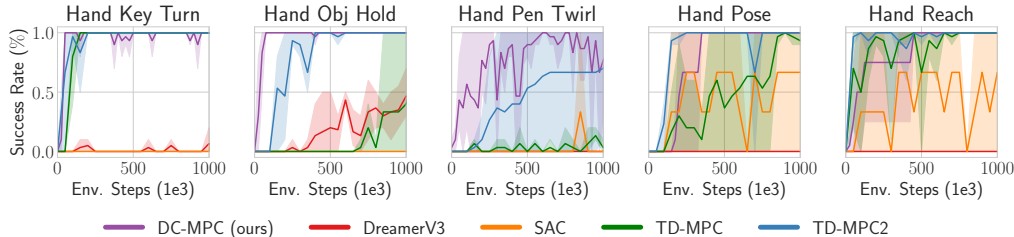

Figure 17: **MyoSuite training curves** We plot the mean (solid line) and the $95\%$ confidence intervals (shaded) across 3 seeds, where each seed averages over 10 evaluation episodes.

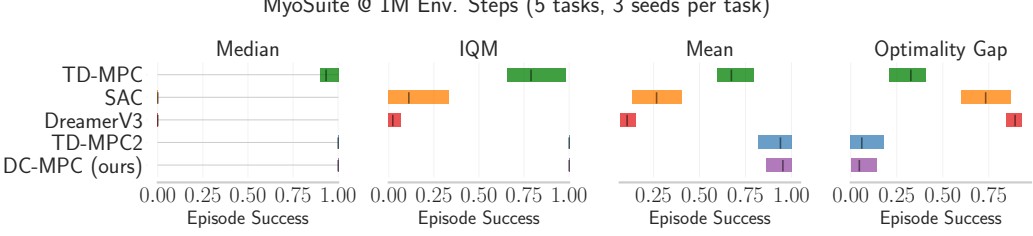

Figure 18: **MyoSuite results** DC-MPC performs similarly to TD-MPC2 in MyoSuite. Error bars represent $95\%$ stratified bootstrap confidence intervals.

–appendices continue on next page–

### B.9 Does DCWM Improve DreamerV3?

In this section, we seek to evaluate what happens when we replace DreamerV3's one-hot discrete encoding with the codebook encoding used in DC-MPC. Fig. 19 shows that in the easy Reacher Hard and Walker Walk environments, FSQ (blue) and one-hot (orange) perform similarly. However, in the difficult Dog Run and Humanoid Walk tasks, no discrete encoding can enable DreamerV3 to perform as well as DC-MPC (purple). We hypothesize that DreamerV3's poor performance in the Dog Run and Humanoid Walk tasks results from its decoder struggling to reconstruct the observations.

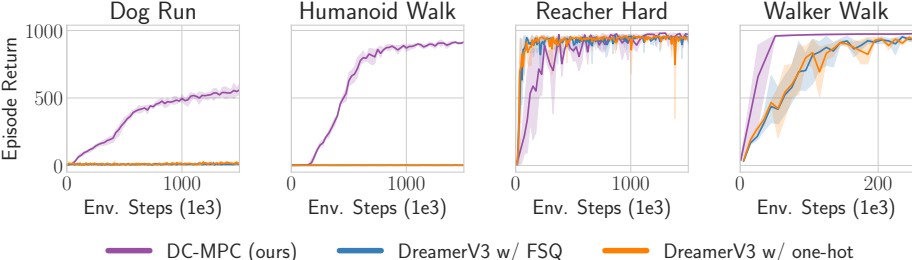

Figure 19: **DreamerV3 with FSQ** Replacing DreamerV3's one-hot encoding (orange) with DC-MPC's codebook encoding (blue) does not improve performance. Moreover, DreamerV3 is not able to learn in the hard Dog Run and Humanoid Walk tasks and is significantly outperformed by DC-MPC (purple).

Learning to minimize the observation reconstruction error has been widely applied in model-based RL (Sutton & Barto, 2018; Ha & Schmidhuber, 2018; Hafner et al., 2019b), and an observation decoder has been a component of many of the most successful RL algorithms to date (Hafner et al., 2023). However, recent work in representation learning for RL (Zhao et al., 2023) and model-based RL (Hansen et al., 2022) has shown that incorporating a reconstruction term into the representation loss can hurt the performance, as learning to reconstruct the observations is inefficient due to the observations containing irrelevant details that are uncontrollable by the agent and do not affect the task.

To provide a thorough analysis of DC-MPC, we include results where we add a reconstruction term to our world model loss in Eq. (8):

$$\mathcal{L}_{\boldsymbol{o}} = \mathbb{E}_{\boldsymbol{o}_t \sim \mathcal{D}}[\|\hat{\boldsymbol{o}}_t - \boldsymbol{o}_t\|_2^2], \quad \hat{\boldsymbol{o}}_t = h_\kappa(\boldsymbol{c}_t), \tag{15}$$

where $h_\kappa$ is a learned observation decoder that takes the latent code as the input and outputs the reconstructed observation. The decoder $h_\kappa$ is a standard MLP. We perform reconstruction at each time step in the horizon. The results in Fig. 20 show that in no environments does reconstruction aid learning, and in some tasks, such as the difficult Dog Run and Humanoid Walk tasks, including the reconstruction term has a significant detrimental effect on the performance, and can even prevent learning completely. Our results support the observations of Zhao et al. (2023) and Hansen et al. (2022) about the lack of need for a reconstruction loss in continuous control tasks. However, it is worth noting that we weighted all loss terms equally whilst the results in Ma et al. (2024) suggest that the observation reconstruction, temporal consistency, and reward prediction loss terms need to be carefully balanced.

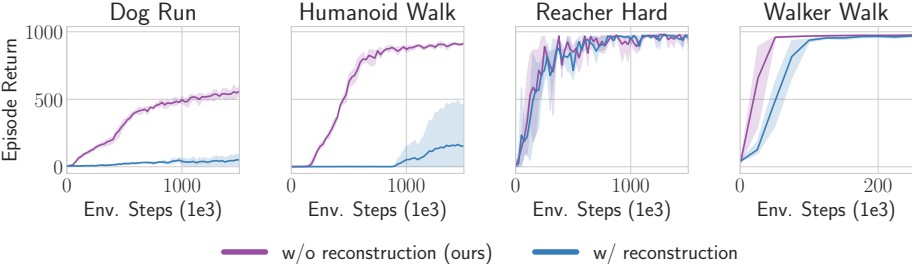

Figure 20: **Reconstruction harms performance** Adding observation reconstruction to DC-MPC (blue) harms the performance of DC-MPC across a mixture of easy and hard DMControl tasks.

### B.10 IMPROVING TD-MPC2 WITH DC-MPC

In this section, we investigate using DC-MPC's latent space inside TD-MPC2. Note that TD-MPC2's latent space is continuous and trained with MSE regression. It also uses simplical normalization (SimNorm) to make its latent space bounded. In these experiments, we removed SimNorm and replaced it with our discrete and stochastic latent space, and then trained using cross-entropy for the consistency loss. In particular, we made the following changes to the TD-MPC2 codebase: *(i)* removed SimNorm, *(ii)* added FSQ to the encoder, *(iii)* modified the dynamics to predict the logits instead of the next latent state, *(iv)* modified the dynamics to use ST Gumbel-softmax sampling for multi-step predictions during training and our weighted average approach during planning, and *(v)* changed the world model's loss coefficients for consistency, value, and, reward, to all be 1.

In Fig. 6, we report aggregate metrics over 3 random seeds in 10 DMControl tasks and 10 Meta-World tasks. Fig. 6 (left) shows the IQM and optimality gap at 1M environment steps over the 20 tasks. It shows that adding DC-MPC's discrete and stochastic latent space to TD-MPC2 offers some improvement. Fig. 6 (right) shows the aggregate training curves (IQM over 10 tasks) for DMControl and Meta-World, respectively. The results show that using DCWM inside TD-MPC2 offers some benefits in the 10 DMControl tasks, whilst in the 10 Meta-World tasks, the performance of all methods seems about equal. This suggests that, in the context of continuous control, discrete and stochastic latent spaces are advantageous for world models. This is an interesting result which we believe motivates further research into discrete and stochastic latent spaces for world models.

## C  IMPLEMENTATION DETAILS

**Architecture**  We implemented DC-MPC with PyTorch (Paszke et al., 2019) and used the AdamW optimizer (Kingma & Ba, 2015) for training the models. All components (encoder, dynamics, reward, actor and critic) are implemented as MLPs. Following Hansen et al. (2023) we let all intermediate layers be linear layers followed by LayerNorm (Ba et al., 2016). We use Mish activation functions throughout. Below we summarize the DC-MPC architecture for our base model.

```
DCMPC(
  (model): WorldModel(
    (_fsq): FSQ(levels=[5, 3])
    (_encoder): ModuleDict(
      (state): Sequential(
        (0): NormedLinear(in_features=obs_dim, out_features=256, act=Mish)
        (1): Linear(in_features=256, out_features=latent_dim*num_channels)
      )
    )
    (_trans): Sequential(
      (0): NormedLinear(in_features=(latent_dim*num_channels)+act_dim, out_features=512, act=Mish)
      (1): NormedLinear(in_features=512, out_features=512, act=Mish)
      (2): Linear(in_features=512, out_features=latent_dim*codebook_size)
    )
    (_reward): Sequential(
      (0): NormedLinear(in_features=(latent_dim*num_channels)+act_dim, out_features=512, act=Mish)
      (1): NormedLinear(in_features=512, out_features=512, act=Mish)
      (2): Linear(in_features=512, out_features=1)
    )
  )
  (_pi): Sequential(
    (0): NormedLinear(in_features=latent_dim*num_channels, out_features=512, act=Mish)
    (1): NormedLinear(in_features=512, out_features=512, act=Mish)
    (2): Linear(in_features=512, out_features=act_dim)
  )
  (_Qs): Vectorized ModuleList(
    (0-4): 5 x Sequential(
      (0): NormedLinear(in_features=(latent_dim*num_channels)+act_dim, out_features=512, act=Mish)
      (1): NormedLinear(in_features=512, out_features=512, act=Mish)
      (2): Linear(in_features=512, out_features=1)
    )
  )
  (_pi_tar): Sequential(
    (0): NormedLinear(in_features=latent_dim*num_channels, out_features=512, act=Mish)
    (1): NormedLinear(in_features=512, out_features=512, act=Mish)
    (2): Linear(in_features=512, out_features=act_dim)
  )
  (Qs_tar): Vectorized ModuleList(
    (0-4): 5 x Sequential(
      (0): NormedLinear(in_features=(latent_dim*num_channels)+act_dim, out_features=512, act=Mish)
      (1): NormedLinear(in_features=512, out_features=512, act=Mish)
      (2): Linear(in_features=512, out_features=1)
    )
  )
)
```

where `obs_dim` is the dimensionality of the observation space, `act_dim` is the dimensionality of the action space, `latent_dim` is the number of the latent dimensions $d$ (default 512), `num_channels` is the number of channels per latent dimension $b$ (default 2), and `codebook_size` is the codebook size $|\mathcal{C}|$ (default 15).

**Statistical significance**  We used five seeds for DC-MPC and three seeds for TD-MPC2/DreamerV3/SAC/TD-MPC in the main figures, at least three seeds for all ablations, and plotted the 95 % confidence intervals as the shaded area, which corresponds to approximately two standard errors of the mean. However, in Figs. 3 and 6 we follow Agarwal et al. (2021) and plot the interquartile mean (IQM) with the shaded area representing $95\%$ stratified bootstrap confidence intervals.

**Hardware**  We used NVIDIA A100s and AMD Instinct MI250X GPUs to run our experiments. All our experiments have been run on a single GPU with a single-digit number of CPU workers.

**Open-source code**  For full details of the implementation, model architectures, and training, please check the code, which is available in the submitted supplementary material and available on github at https://github.com/aidanscannell/dcmpc.

–appendices continue on next page–

**Hyperparameters**  Table 1 lists all of the hyperparameters for training DC-MPC which were used for the main experiments and the ablations.

Table 1: **DC-MPC hyperparameters** We kept most hyperparameters fixed across tasks. However, we set task-specific exploration noise schedules and $N$-step returns.

| HYPERPARAMETER | VALUE | DESCRIPTION |
|---|---|---|
| **TRAINING** | | |
| ACTION REPEAT | 2 (1 IN MYOSUITE) | |
| MAX EPISODE LENGTH ($T$) | 500 IN DMCONTROL | ACTION REPEAT MAKES THIS 1000 |
| | 100 IN META-WORLD | ACTION REPEAT MAKES THIS 200 |
| | 100 IN MYOSUITE | |
| NUM. EVAL EPISODES | 10 | |
| RANDOM EPISODES ($N_{\text{RANDOM EPISODES}}$) | 10 | NUM. RANDOM EPISODES AT START |
| **MPPI PLANNING** | | |
| PLANNING HORIZON | 3 | |
| PLANNING ITERATIONS ($J$) | 6 | |
| POPULATION SIZE ($N_p$) | 512 | |
| PRIOR POPULATION SIZE ($N_\pi$) | 24 | NUM. POLICY SAMPLES TO WARM START |
| NUMBER OF ELITES ($K$) | 64 | |
| MINIMUM STD | 0.05 | |
| MAXIMUM STD | 2 | |
| (INVERSE) TEMPERATURE ($\tau_{\text{MPPI}}$) | 0.5 | |
| **TD3** | | |
| ACTOR UPDATE FREQ. | 2 | UPDATE ACTOR LESS THAN CRITIC |
| BATCH SIZE | 512 | |
| BUFFER SIZE | $10^6$ | |
| DISCOUNT FACTOR ($\gamma$) | 0.99 | |
| EXPLORATION NOISE | $\text{Linear}(1.0, 0.1, 50)$ (EASY) | DMCONTROL |
| | $\text{Linear}(1.0, 0.1, 150)$ (MEDIUM) | DMCONTROL |
| | $\text{Linear}(1.0, 0.1, 500)$ (HARD) | DMCONTROL |
| | $\text{Linear}(1.0, 0.1, 250)$ | META-WORLD & MYOSUITE |
| LEARNING RATE | $3 \times 10^{-4}$ | |
| MLP DIMS | $[512, 512]$ | FOR ACTOR/CRITIC/DYNAMICS/REWARD |
| MOMENTUM COEF. ($\tau$) | 0.005 | |
| NUM. $Q$-FUNCTIONS ($N_q$) | 5 | |
| NUM. $Q$-FUNCTIONS TO SAMPLE | 2 | |
| NOISE CLIP ($c$) | 0.3 | |
| N-STEP TD | 1 OR 3 IN DMCONTROL | |
| | 3 IN META-WORLD | |
| | 1 OR 5 IN MYOSUITE | |
| POLICY NOISE | 0.2 | |
| UPDATE-TO-DATA (UTD) RATIO | 1 | |
| **WORLD MODEL** | | |
| DISCOUNT FACTOR ($\gamma$) | 0.9 | |
| ENCODER LEARNING RATE | $10^{-4}$ | |
| ENCODER MLP DIMS | $[256]$ | |
| FSQ LEVELS | $[5, 3]$ | GIVES $|\mathcal{C}| = 5 \times 3 = 15 \approx 2^4$ |
| HORIZON ($H$) | 5 | FOR WORLD MODEL TRAINING |
| LATENT DIMENSION ($d$) | 512 | |
| | 1024 (HUMANOID/DOG) | |

–appendices continue on next page–

## D  BASELINES

In this section, we provide further details of the baselines we compare against.

- **DreamerV3 (Hafner et al., 2023)** is a reinforcement learning algorithm that uses a world model to predict outcomes, a critic to judge their value, and an actor to choose actions to maximize value. It uses symlog loss for training and operates on model states from imagination data. The critic is a categorical distribution with exponentially spaced bins, and the actor is trained with entropy regularization and return normalization. The world model is only used for training and there is no decision-time planning. In contrast, DC-MPC learns a deterministic encoder with a discrete latent space and stochastic dynamics in the world model. We report the results of DreamerV3 from the TD-MPC2 official repository [2].

- **Temporal Difference Model Predictive Control 2 (TD-MPC2, Hansen et al. (2023))** is a decoder-free model-based reinforcement learning algorithm with a focus on scalability and sample efficiency. It includes an encoder, latent transition dynamics, a reward predictor, a terminal value (critic), and a policy prior (actor). In contrast to DreamerV3, it utilizes a deterministic encoder and transition dynamics implemented with MLPs, layer normalization (Ba et al., 2016) and Mish (Misra, 2019) activations. To avoid exploding gradients and representation collapse, the latent space is normalized with projection followed by a softmax operation. All components except the policy prior are trained jointly based on predicting the latent embedding, reward prediction, and value prediction, while reward and value predictions are based on discrete regression in log-transformed space. Similarly, we use a deterministic encoder, but we train the transition dynamics with a cross-entropy loss function, which considers multi-modality and uncertainties, and we decouple representation learning from value learning. We report the results from the TD-MPC2 official repository [3].

- **Temporal Difference Model Predictive Control (TD-MPC, Hansen et al. (2022))** is the first version of TD-MPC2. It is also a decoder-free model-based RL algorithm consisting of an encoder, latent transition dynamics, reward predictor, terminal value (critic), and policy prior (actor). In contrast to TD-MPC2, it does not apply simplical normalization (SimNorm) to its latent state, it trains the reward and value prediction using the MSE loss instead of the cross-entropy loss, and it uses SAC as the underlying RL algorithm. We refer the reader to the TD-MPC paper for further details. We report the results from the TD-MPC2 official repository [4].

- **Soft Actor-Critic (SAC, Haarnoja et al. (2018)** is an off-policy model-free RL algorithm based on the maximum entropy RL framework. That is, it attempts to succeed at the task whilst acting as randomly as possible. It is worth highlighting that TD-MPC2 uses SAC as it's underlying model-free RL algorithm. We report the results from the TD-MPC2 official repository [5].

–appendices continue on next page–

---

[2] https://github.com/nicklashansen/tdmpc2/tree/main/results/dreamerv3
[3] https://github.com/nicklashansen/tdmpc2/tree/main/results/tdmpc2
[4] https://github.com/nicklashansen/tdmpc2/tree/main/results/tdmpc
[5] https://github.com/nicklashansen/tdmpc2/tree/main/results/sac

# E TASKS

We evaluate our method in 30 tasks from the DeepMind Control suite (Tassa et al., 2018), 45 tasks from Meta-World (Yu et al., 2019) and 5 tasks from MyoSuite (Vittorio et al., 2022). Tables 2 to 4 provide details of the environments we used, including the dimensionality of the observation and action spaces.

Table 2: **DMControl** We consider a total of 30 continuous control tasks from DMControl.

| TASK | OBSERVATION DIM | ACTION DIM | SPARSE? |
|------|:---:|:---:|:---:|
| ACROBOT SWINGUP | 6 | 1 | N |
| CARTPOLE BALANCE | 5 | 1 | N |
| CARPOLE BALANCE SPARSE | 5 | 1 | Y |
| CARTPOLE SWINGUP | 5 | 1 | N |
| CARTPOLE SWINGUP SPARSE | 5 | 1 | Y |
| CHEETAH RUN | 17 | 6 | N |
| CUP CATCH | 8 | 2 | Y |
| CUP SPIN | 8 | 2 | N |
| DOG RUN | 223 | 38 | N |
| DOG STAND | 223 | 38 | N |
| DOG TROT | 223 | 38 | N |
| DOG WALK | 223 | 38 | N |
| FINGER SPIN | 9 | 2 | Y |
| FINGER TURN EASY | 12 | 2 | Y |
| FINGER TURN HARD | 12 | 2 | Y |
| FISH SWIM | 24 | 5 | N |
| HOPPER HOP | 15 | 4 | N |
| HOPPER STAND | 15 | 4 | N |
| HUMANOID RUN | 67 | 24 | N |
| HUMANOID STAND | 67 | 24 | N |
| HUMANOID WALK | 67 | 24 | N |
| PENDULUM SPIN | 3 | 1 | N |
| PENDULUM SWINGUP | 3 | 1 | N |
| QUADRUPED RUN | 78 | 12 | N |
| QUADRUPED WALK | 78 | 12 | N |
| REACHER EASY | 6 | 2 | Y |
| REACHER HARD | 6 | 2 | Y |
| WALKER RUN | 24 | 6 | N |
| WALKER STAND | 24 | 6 | N |
| WALKER WALK | 24 | 6 | N |

–appendices continue on next page–

Table 3: **Meta-World** We consider a total of 45 continuous control tasks from Meta-World. This benchmark is designed for multitask research so all tasks share similar embodiment, observation space, and action space.

| TASK | OBSERVATION DIM | ACTION DIM | SPARSE? |
|---|---|---|---|
| ASSEMBLY | 39 | 4 | N |
| BASKETBALL | 39 | 4 | N |
| BIN PICKING | 39 | 4 | N |
| BOX CLOSE | 39 | 4 | N |
| BUTTON PRESS | 39 | 4 | N |
| BUTTON PRESS TOPDOWN | 39 | 4 | N |
| BUTTON PRESS TOPDOWN WALL | 39 | 4 | N |
| BUTTON PRESS WALL | 39 | 4 | N |
| COFFEE BUTTON | 39 | 4 | N |
| COFFEE PUSH | 39 | 4 | N |
| COFFEE PULL | 39 | 4 | N |
| DIAL TURN | 39 | 4 | N |
| DISASSEMBLE | 39 | 4 | N |
| DOOR CLOSE | 39 | 4 | N |
| DOOR LOCK | 39 | 4 | N |
| DOOR OPEN | 39 | 4 | N |
| DOOR UNLOCK | 39 | 4 | N |
| DRAWER CLOSE | 39 | 4 | N |
| FAUCET CLOSE | 39 | 4 | N |
| FAUCET OPEN | 39 | 4 | N |
| HAMMER | 39 | 4 | N |
| HAND INSERT | 39 | 4 | N |
| HANDLE PRESS | 39 | 4 | N |
| HANDLE PRESS SIDE | 39 | 4 | N |
| HANDLE PULL | 39 | 4 | N |
| HANDLE PULL SIDE | 39 | 4 | N |
| LEVER PULL | 39 | 4 | N |
| PEG INSERT SIDE | 39 | 4 | N |
| PEG UNPLUG SIDE | 39 | 4 | N |
| PICK OUT OF HOLE | 39 | 4 | N |
| PICK PLACE | 39 | 4 | N |
| PLATE SLIDE | 39 | 4 | N |
| PLATE SLIDE BACK | 39 | 4 | N |
| PLATE SLIDE BACK SIDE | 39 | 4 | N |
| PLATE SLIDE SIDE | 39 | 4 | N |
| PUSH | 39 | 4 | N |
| PUSH WALL | 39 | 4 | N |
| REACH WALL | 39 | 4 | N |
| SOCCER | 39 | 4 | N |
| STICK PULL | 39 | 4 | N |
| STICK PUSH | 39 | 4 | N |
| SWEEP | 39 | 4 | N |
| SWEEP INTO | 39 | 4 | N |
| WINDOW CLOSE | 39 | 4 | N |
| WINDOW OPEN | 39 | 4 | N |

Table 4: **MyoSuite** We consider a total of 5 continuous control tasks from MyoSuite. This benchmark is designed for high-dimensional muscoloskeletal motor control which involves complex object manipulation with a dexterous hand.

| TASK | OBSERVATION DIM | ACTION DIM | SPARSE? |
|---|---|---|---|
| KEY TURN | 93 | 39 | N |
| OBJECT HOLD | 91 | 39 | N |
| PEN TWIRL | 83 | 39 | N |
| POSE | 108 | 39 | N |
| REACH | 115 | 39 | N |

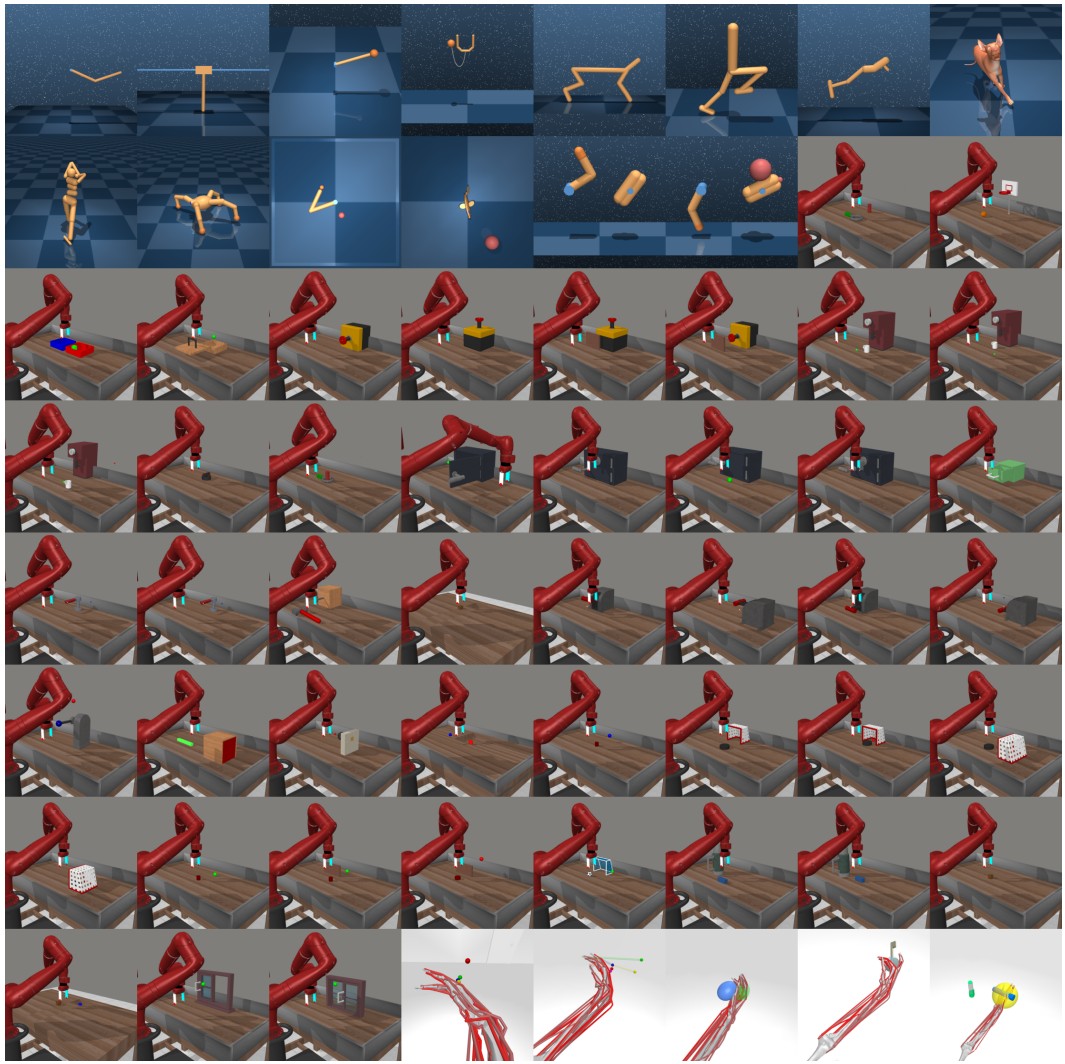

Figure 21: **Tasks visualizations** Visualization of the DMControl, Meta-World, and MyoSuite tasks used throughout the paper.

