# OpenReview forum: "Discrete Codebook World Models for Continuous Control"
_ICLR.cc/2025/Conference — ICLR 2025 Poster_

### Official Review · Reviewer_iQtr · 2024-10-26

**Soundness:** 4
**Presentation:** 3
**Contribution:** 3
**Rating:** 6
**Confidence:** 5

**Summary:**

This paper explores the application of discrete latent spaces for continuous control in reinforcement learning (RL), an innovative perspective other than the typical use of continuous latent spaces. The authors introduce the Discrete Codebook World Model (DCWM), which utilizes quantized discrete codebook encodings to represent latent states. The DCWM is demonstrated to surpass recent state-of-the-art RL algorithms, including TD-MPC2 and DreamerV3, especially in those tasks with high-dimensional action spaces.

**Strengths:**

1. Although similar ideas like the token-based world model were recently published, the proposed method seems computationally efficient, benefiting from the 'Finite Scalar Quantization' paper.
2. The paper first made the discrete latent space method work in continuous control tasks.
3. The paper writing is clear and easy to understand.

**Weaknesses:**

1. This paper lacks comparisons with the more advanced baselines like EfficientZero-V2.
2. The authors didn't test the proposed method with visual inputs, which made this paper not that strong.

**Questions:**

1. What's your idea about if FSQ-based discretization could be used for action space discretization? (like action sampling) If yes, how will you do that?
2. Although it could be more computationally challenging, could you please provide several visual input results? You can ignore the most difficult tasks like humanoid walk/run.

---

> ### Author Response · Authors · 2024-11-25
>
> We thank Reviewer iQtr for their feedback. We refer the reviewer to our general response where we outline the improvements to our submission. Here we address specific questions.
>
> **W1: This paper lacks comparisons with the more advanced baselines like EfficientZero-V2.**
>
> **AW1:** EfficientZero-V2 proposes improvements on the policy and planning level:
> > We design two key algorithmic enhancements: a sampled-based tree search for action planning, ensuring policy improvement in continuous action spaces, and a search-based value estimation strategy to more efficiently utilize previously gathered data and mitigate off-policy issues.
>
> In contrast, our paper investigates and proposes changes to the latent embedding and dynamics functions. While it stands to reason that many methods improving upon DreamerV3 and TD-MPC2, such as EfficientZero-V2, can also benefit from our proposed changes, we are unfortunately not able to compare against all possible variations of DreamerV3, which is cited 471 times.
> Hence, we specifically compare against the baseline implementations of DreamerV3 and TD-MPC2 which use different latent embeddings (continuous vs discrete), as well as perform ablation studies into the different discretization schemes of the latent embedding space. We agree with you that it would be beneficial to develop a kitchen sink approach combining these advancements from different papers in future work, but we think this is out of scope for this paper.
> However, please see our updated draft for further ablation studies and additional experiments.
>
> **Q1: What's your idea about if FSQ-based discretization could be used for action space discretization? (like action sampling) If yes, how will you do that?**
>
> **AQ1:** In this paper we are specifically interested in applying FSQ to the discretization of latent state/observation embeddings. We did not investigate the possibility of action space discretization and leave this to future work.

---

> > ### Comment · Reviewer_iQtr · 2024-11-25
> >
> > It seems that the author didn't provide results with visual inputs. I will maintain my scores.

---

### Official Review · Reviewer_HZL5 · 2024-11-03

**Soundness:** 3
**Presentation:** 3
**Contribution:** 3
**Rating:** 8
**Confidence:** 5

**Summary:**

This paper proposes a discrete latent space approach for world models in continuous control, utilizing FSQ within a MBRL algorithm, DCWM, following TD-MPC2. The experiments, conducted on DMC and Metaworld, demonstrate improvements over state-of-the-art methods, including TD-MPC2 and DreamerV3. Additionally, the authors analyze the effectiveness of discrete vs. continuous latents, classification vs. regression objectives, and deterministic vs. stochastic transitions.

**Strengths:**

1. The investigation into latent space design is valuable for the model-based RL community.
2. The experiments are informative, offering insights across multiple dimensions of the design.

**Weaknesses:**

1. A significant concern is the need for clarification; some claims may be overstated. See questions below.
2. For an impactful empirical study on algorithmic design, the scale of experiments—particularly in terms of the number of environments and tasks—seems insufficient. The experimental design is also somewhat confusing; see questions below.

**Questions:**

On clarifcation (major):

1. The claim in the abstract that discrete latent spaces typically underperform in continuous control tasks needs specification. First of all, to my knowledge, TD-MPC2 only conducts experiments on state-based tasks, lacking results on pixel-based continuous control, while DreamerV3 does include these. Second, it is unfair to compare methods with (TD-MPC) and without inference-time planning (Dreamer), as in the MuZero paper we can see that this makes a great performance boost. I think this work can naturally motivate itself from TD-MPC2 alone instead of comparing DreamerV3 and TD-MPC2 without head-to-head experiments.
2. A preliminary section clarifying different discrete encodings is necessary; I find the distinction between label encoding and one-hot encoding unclear for the first-time reading. The methods section extensively describes FSQ, which is not originally proposed here.
3. Also, the methods section contains many elements that directly follow TD-MPC2, which may confuse non-expert readers regarding what is novel. Clear differentiation is needed. For example, TD-MPC2 exactly uses an ensemble of five critics with randomly sampled two for bootstrap, but this is not explicitly mentioned by the authors in Line 277.
4. Algorithm 1 appears to be a standard procedure in model-based RL and could be moved to the appendix.

On clarifcation (minor):

1. Can you elaborate on what you mean by "codebook encoding captures similarity between observations" and how this contrasts with one-hot encodings in Line 248?
2. I did not understand why "we did not conﬁgure the transition model to accept the label and one-hot encodings as this resulted in the agent being unable to learn." Can you explain more?
3. I did not understand the sentence "our codebook is conﬁgured with b = 2 channels and the label encoding incorrectly assumes an ordinal structure through both." How does the hyperparameter $b$ in your proposed codebook encoding affect label encoding?
4. In Line 514, why is $|C|=2^4$ for $L=[5,3]$, when it is claimed $|C|=\prod L_i$ in Line 224.

On experiments:

1. Could you provide additional experiments on more tasks, such as ManiSkill2/Myosuite, which TD-MPC2 used? It would also be beneficial to include experiments on more base algorithms, such as DreamerV3 (see question 2).
2. The most direct evidence that the proposed discrete latents outperform one-hot and continuous latents would be to replace the latent space in DreamerV3 and TD-MPC2 with your proposed approach. Why wasn't this head-to-head comparison conducted?
3. Have you compared FSQ and VQ (referred to as dictionary learning in Line 206)?
4. Codebook encoding only outperforms one-hot encoding in one task in Figure 6, yet it is claimed to be more computationally efficient in Line 485. Is there any numerical analysis to support this claim?

Overall, I find this paper interesting. If the authors adequately address my concerns, I would be willing to significantly improve my rating.

---

> ### Comment · Reviewer_HZL5 · 2024-11-25
>
> Since no rebuttal is provided, I decided to keep my score currently.

---

> > ### Comment · Reviewer_Nctn · 2024-11-25
> >
> > For what it's worth, I think these are important questions to address! I think Reviewer HZL5 and I might disagree about whether they should significantly impede acceptance of this work, but I do think these if these were addressed, the paper would be clearer. I think Clarifications (Major) 2, and Experiments 2 are especially interesting to address. It appears also that it would be good to clarify the discussion in Appendix D with respect to the latent space differences in greater detail.

---

> ### Author Response · Authors · 2024-11-25
>
> We are glad to see that you found our paper insightful and experiments informative. Please, see below our answers to your raised questions:
>
> **Q1: The claim in the abstract that discrete latent spaces typically underperform in continuous control tasks needs specification...**
>
> **AQ1:** We thank the reviewer for this comment. First, we would like to point out that we compare our approach to both DreamerV3 and TD-MPC2 as baselines as they utilize world models with different latent state embeddings. Furthermore, a sole comparison between TD-MPC2 to DreamerV3 was already presented by [Hansen et al., 2024], thus, we found it informative and beneficial for the reader to perform a comparison of our presented approach to both methods, especially due to their use of different encoding schemes. We have clarified our statement about the loss of performance when using discrete latent spaces - we did not mean to downplay DreamerV3's achievements unfairly. As you can see in Figures 5/12 and Line 462, we performed an in-depth evaluation of different latent spaces and found the proposed discrete latent space to perform better than continuous latent spaces, even though it stands to reason that discrete codes may lead to a decrease in performance due to information loss.
>
> **Q2: A preliminary section clarifying different discrete encodings is necessary; I find the distinction between label encoding and one-hot encoding unclear for the first-time reading.**
>
> **AQ2:** Thank you for bringing this to our attention. We have now added a Preliminaries section (Section 3) which introduces and compares one-hot, label, and codebook encodings. We think this is a nice addition to the paper but please let us know if anything remains unclear.
>
> **Q3: The methods section extensively describes FSQ, which is not originally proposed here. Also, the methods section contains many elements that directly follow TD-MPC2, which may confuse non-expert readers regarding what is novel. Clear differentiation is needed. For example, TD-MPC2 exactly uses an ensemble of five critics with randomly sampled two for bootstrap, but this is not explicitly mentioned by the authors in Line 277.**
>
> **AQ3:** We clearly indicate and reference to the reader that FSQ is prior work, please see Line 208. As the discretization scheme is an important aspect of the paper, we find it appropriate to discuss its background and intuition in more detail. Nevertheless, the reviewer is correct that we intentionally kept aspects of our algorithm similar to TD-MPC2 in an attempt to isolate the impact of our world model's latent space. More specifically, we used an ensemble of five critics, the same NN architectures, and we used MPPI for decision-time planning. To improve clarity, we added a further discussion on the specific contributions and differentiations to other model-based RL methods in Section 4.1.
>
> **Q4: Can you elaborate on what you mean by "codebook encoding captures similarity between observations" and how this contrasts with one-hot encodings in Line 248?**
>
> **AQ4:** We thank the reviewer for raising this question as this is an important point that we want to make sure is clear in our paper. We have added further clarification regarding this statement in the new "Preliminaries" section (Section 3) which we include here for convenience:
> > - **One-hot encoding** Given categories $A$, $B$, $C$, and $D$, a one-hot encoding would take the form $A=[1,0,0,0]$, $B=[0,1,0,0]$, $C=[0,0,1,0]$ and $D=[0,0,0,1]$, respectively. This encoding creates equidistant vectors in the feature space, as the Euclidean distance between any two one-hot vectors is always $\sqrt{2}$.
> > - **Label encoding** Given categories $A$, $B$, $C$, and $D$, label encoding would result in $A=1$, $B=2$, $C=3$, and $D=4$ respectively.
> > - **Codebook encoding** Given categories $A$, $B$, $C$, and $D$, a codebook might encode them as $A=[0.5,0.5]$, $B=[0.5,-0.5]$, $C=[-0.5,0.5]$, and $D=[-0.5,-0.5]$ respectively.
>
> >Mathematically, if we have an ordinal relationship $A < B < C$, label and codebook encodings can ensure $|e(A) - e(B)| < |e(A) - e(C)|$, where $e(x)$ is the encoding function. One-hot encoding, however, results in $|e(A) - e(B)| = |e(A) - e(C)| = \sqrt{2}$ for all distinct pairs, eliminating any notion of ordering. Whilst this may be beneficial in some scenarios, e.g. when modelling distinct categories like fruits, it means that they cannot capture the inherent ordering in continuous data. In contrast, label and codebook encodings can capture ordinal relationships.

---

> ### Author Response · Authors · 2024-11-25
>
> **Q5: I did not understand why "we did not conﬁgure the transition model to accept the label and one-hot encodings as this resulted in the agent being unable to learn."**
>
> **AQ5:** This is an important point so thank you for highlighting that this is not clear.  We have updated Section 5.3 to provide more clarity.
>
> > Our world model consists of NNs for the dynamics $P_{\phi}(\mathbf{c}' \mid \mathbf{c}, \mathbf{a})$, reward $R_{\xi}(\mathbf{c}, \mathbf{a})$, critic $Q_{\psi}(\mathbf{c}, \mathbf{a})$, and prior policy $\pi\_{\eta}(\mathbf{c})$, which all make predictions given the discrete codebook encoding $\mathbf{c}=\mathbf{e}\_{\text{codes}}$. We compare DCWM's codebook encoding $\mathbf{e}\_{\text{codes}}=\mathbf{c}^{i} \in \mathcal{C}$ to (i) label encoding $e\_{\text{labels}} = i \in \{1,\dots|\mathcal{C}|\}$ and (ii) one-hot encoding $\mathbf{e}\_{\text{one-hot}} =\mathbf{v} \in \{0,1\}^{|\mathcal{C}|}$ given $\textstyle \sum_{i=1}^{|\mathcal{C}|} v_{i} =1$. In Fig. 6, we evaluate what happens when we replace the codebook encoding $\mathbf{c}$ with either one-hot $\mathbf{e}\_{\text{one-hot}}$ or label $\mathbf{e}\_{\text{label}}$ encodings. In these experiments, we did not modify the dynamics $P\_{\phi}(\mathbf{c}' \mid \mathbf{c}, \mathbf{a})$, that is, the dynamics continued to make predictions using the codebook encoding $\mathbf{c}$ and did not use the one-hot or label encodings. This is because we found that using one-hot and label encodings for the dynamics led to the agent being unable to learn. This suggests that our codebook encoding is essential when leveraging a discrete latent space for learning continuous control. More specifically, we evaluated the following experiment configurations:
>  > 1. **Codes (ours)**: All components used codes: $P\_{\phi}(\mathbf{c}' \mid \mathbf{c}, \mathbf{a})$, reward $R_{\xi}(\mathbf{c}, \mathbf{a})$, critic $Q\_{\psi}(\mathbf{c}, \mathbf{a})$ and prior policy $\pi_{\eta}(\mathbf{c})$.
>
> > 2. **Labels**: Dynamics model used codes $P_{\phi}(\mathbf{c}' \mid \mathbf{c}, \mathbf{a})$ whilst reward $R_{\xi}(\mathbf{e}\_{\text{labels}}, \mathbf{a})$, critic $Q_{\psi}(\mathbf{e}\_{\text{labels}}, \mathbf{a})$ and prior policy $\pi_{\eta}(\mathbf{e}\_{\text{labels}})$ used labels $\mathbf{e}\_{\text{labels}}$ obtained from the code's index $i$ in the codebook.
>
> > 3. **One-hot**: Dynamics model used codes $P_{\phi}(\mathbf{c}' \mid \mathbf{c}, \mathbf{a})$ whilst reward $R_{\xi}(\mathbf{e}\_{\text{one-hot}}, \mathbf{a})$, critic $Q_{\psi}(\mathbf{e}\_{\text{one-hot}}, \mathbf{a})$ and prior policy $\pi_{\eta}(\mathbf{e}\_{\text{one-hot}})$, used one-hot $\mathbf{e}\_{\text{one-hot}}$ obtained by applying PyTorch's `torch.nn.functional.one_hot` to the label encoding.
>
> Please let us know if anything remains unclear or if you have any suggestions for improvement.
>
> **Q6: I did not understand the sentence "our codebook is conﬁgured with b = 2 channels and the label encoding incorrectly assumes an ordinal structure through both." How does the hyperparameter $b$ in your proposed codebook encoding affect label encoding?**
>
> **AQ6:** We would like to refer you to the new "Preliminaries" section (Section 3) where we discuss one-hot, label, and codebook encodings, and how they imply ordinal structure. To avoid this confusion in the future, we have also added the following text to Section 5.3:
> > The label encoding (blue) struggles to learn in the Humanoid Walk task and is often less sample efficient than the alternative encodings. This is likely because the label encoding is not expressive enough to model the multi-dimensional ordinal structure of our codebook. Let us provide intuition via a simple example. Our codebook has $b=2$ channels, so two different codes may take the form $e\_{\text{codes}}(A)=[0.5, -0.5]$ and $e\_{\textbf{codes}}(B)=[0, 0.5]$. As a result, our codebook encoding can model ordinal structure in both of its channels, i.e., $e\_{\text{codes}}(A)_1>e\_{\text{codes}}(B)_1$ whilst $e\_{\text{codes}}(A)_2 < e\_{\text{codes}}(B)_2$. The corresponding label encoding would encode this as $e\_{\text{labels}}(A)=1$ and $e\_{\text{labels}}(B)=2$, which incorrectly implies that $B>A$. In short, the label encoding cannot model the multi-dimensional ordinal structure of the codebook $\mathcal{C}$.

---

> ### Author Response · Authors · 2024-11-25
>
> **Q7: In Line 514, why is $|\mathcal{C}|=2^4$ for $L=[5,3]$, when it is claimed $|\mathcal{C}| = \prod L_i$ in Line 224.**
>
> **AQ7:** We invite the reviewer to check Line 514 again (now line 516), which reads as $|\mathcal{C}|\approx 2^4$ and not as $|\mathcal{C}| = 2^4$. Data types are often written as $2^X$ because of the binary nature of computer systems. For example, real numbers are often represented in single precision (32 bits) which are of size $2^{32}$. As such, representing the size of discrete codebooks using $2^X$ makes it easier to compare sizes, which is why it is common in FSQ and VQ-VAE to represent the codebook size as $|\mathcal{C}| \approx 2^X$. We note that the reviewer is correct that the codebook size is given by $|\mathcal{C}|=5 \times 3 =15 \approx 2^4$. In order to prevent this confusion in the future, we have updated Line 516 to read as:
> > We find that a latent dimension of $d=1024$ with FSQ levels $\mathcal{L}=[5,3]$, which corresponds to a codebook size $|\mathcal{C}| = 5 \times 3 = 15 \approx2^{4}$, performs best in the harder DMC tasks.
>
> **Q8: Could you provide additional experiments on more tasks, such as ManiSkill2/Myosuite, which TD-MPC2 used?**
>
> **AQ8:** We have been able to get results in 45 MetaWorld tasks (previously we only had 7) and 5 MyoSuite tasks. We have compared our method with baselines on Myosuite tasks in Section B.10 and Figure 19 and Figure 20. Please also see our general response.
>
> **Q9: It would also be beneficial to include experiments on more base algorithms, such as DreamerV3 (see question 2).**
>
> **AQ9:** We have included results for soft actor-critic (SAC) and the original TD-MPC in all DMControl and Meta-World tasks. As such, we are now comparing against TD-MPC, TD-MPC2, SAC, and DreamerV3, in all figures that report baselines.
>
> **Q10: The most direct evidence that the proposed discrete latents outperform one-hot and continuous latents would be to replace the latent space in DreamerV3 and TD-MPC2 with your proposed approach. Why wasn't this head-to-head comparison conducted?**
>
> **AQ10:** Please see Appendix B.9 (and our general response) where we have included a section comparing DCWM and DreamerV3.
>
> **Q11: Have you compared FSQ and VQ (referred to as dictionary learning in Line 206)?**
>
> **AQ11:** We agree this is an interesting comparison and thank the reviewer for raising it. In response to this question, we have included results for VQ-VAE to Appendix B.7. Please see our response to all reviewers for further details.
>
> **Q12: Codebook encoding only outperforms one-hot encoding in one task in Figure 6, yet it is claimed to be more computationally efficient in Line 485. Is there any numerical analysis to support this claim?**
>
> **AQ12:** One-hot, label, and codebook encodings have different impacts on computational efficiency in neural networks. First, one-hot encodings increase the input dimension as they create new columns for each category. In our setting, we have $d=512$ latent dimensions and $15$ discrete values so the one-hot encoding is of size $512\times15=7680$. In contrast, label encodings are of size $d=512$. Codebook encodings offer a trade-off as they are off size $d \times b = 512 \times 2 =1024$. Second, one-hot encodings create sparse data as there are not many non-zero values in the encoding. In contrast, both label and codebook encodings create dense data as they have many non-zero values. Due to the increased dimensionality and sparsity, one-hot encoding may lead to longer training times compared to label encoding. The table below summarizes the differences between the encodings. We have also updated Figure 7 (now Figure 6) to compare both sample efficiency and computational efficiency of one-hot, label, and codebook encodings in the hard Dog Run and Humanoid Walk tasks. It shows that one-hot encodings result in slower training.
>
> | Aspect | One-Hot Encoding | Label Encoding | Codebook Encoding |
> |--------|------------------|----------------|-------------------|
> | Dimensionality | Increases significantly $d \times \|\mathcal{C}\|=512\times 15=7680$ | Maintains original $d=512$ | Increases a little $d \times b = 512 \times 2 = 1024$ |
> | Sparsity | Creates sparse data (not many non-zero values) | Creates dense data (many non-zero values) |  Creates dense data (many non-zero values) |
> | Memory usage | Higher | Lower | Lower
> | Training time | Generally longer | Generally shorter | Generally shorter |
> | Computational complexity | Higher | Lower | Lower |
> | Ordinal relationship | No implied order | Implies ordinal relationship | Implies ordinal relationship |

---

> > ### Comment · Reviewer_HZL5 · 2024-11-25
> >
> > I thank the authors for their great efforts to address my questions. I promise to carefully check the response in the final days of the rebuttal period.

---

> > > ### Author Response · Authors · 2024-11-25
> > >
> > > Thank you for letting us know. This comment is really appreciated as we know we have posted the rebuttal a bit late (it took quite some time to run the experiments).

---

> > > > ### Comment · Reviewer_HZL5 · 2024-11-29
> > > >
> > > > Thanks again for the detailed response. I have reviewed the rebuttal alongside the revised manuscript and would like to provide the following comments:
> > > >
> > > > **Ideally Addressed:**
> > > >
> > > > - Q3, 6, 7 (sorry for my stupid arithmetic mistake), 11, 12: These points have been satisfactorily addressed. I appreciate the detailed clarifications regarding different encoding methods and the extensive additional experiments.
> > > >
> > > > **Partially Addressed and Requiring Further Clarification:**
> > > >
> > > > - AQ1: My concern is not about the superiority of DCWM over DreamerV3. Rather, it is about the inaccurate claim in the manuscript that "discrete latent spaces typically underperform in continuous control tasks." While this may hold for state-based control tasks (as suggested by the experimental results of TD-MPC2), the comparison is not head-to-head due to significant implementation differences between DreamerV3 and TD-MPC2. Moreover, TD-MPC2 does not evaluate visual control tasks, whereas DreamerV3 does. I recommend revising this claim to make it more precise and specific, as it forms a key motivation for the work.
> > > > - AQ2 & AQ4: I disagree with the statement that codebook encodings can ensure a complete ordering. Actually in the example provide, we have $|e(A)-e(B)|=|e(A)-e(C)|$.
> > > > - AQ5: I still don't understand the literal meaning of "using one-hot and label encodings for the dynamics led to the agent being unable to learn"? Does "unable to learn" mean that the empirical performance is poor?
> > > > - AQ8: Since both TD-MPC2 and DCWM achieve a final success rate of nearly 100%, I recommend plotting the IQM of learning curves (like Fig 4 in TD-MPC2) instead of the final performance. Note that I am not complaining about the performance but solely about the presentation.
> > > > - AQ9: Apologies for my earlier unclear comment. My suggestion was specifically regarding FSQ combined with a base method like DreamerV3 (as mentioned in Q10). I am not aiming to add numerous baselines solely for comparison.
> > > > - AQ10:
> > > >   - Appendix B.9: The finding that reconstruction harms performance is intriguing. Some preliminary literature [1] has explored the role of the reconstruction component in world models. I believe the weight of the reconstruction loss could play a significant role here. To clarify, I am not requesting additional experiments but hope this observation inspires the authors.
> > > >   - Appendix B.11: I am pleased to see that TD-MPC2 itself benefits from a discrete latent space. These results suggest that the superior performance of DCWM is also influenced by implementation differences compared to TD-MPC2. I recommend highlighting this point more explicitly in future revisions.
> > > >
> > > > I have currently updated my score to 5 and look forward to further discussions with the authors.
> > > >
> > > > [1] HarmonyDream: Task Harmonization Inside World Models. ICML 24.

---

> > > > > ### Author Response · Authors · 2024-11-29
> > > > >
> > > > > Thank you for your detailed response. We now address your questions requiring further clarification:
> > > > >
> > > > > **AQ1: My concern is not about the superiority of DCWM over DreamerV3. Rather, it is about the inaccurate claim in the manuscript that "discrete latent spaces typically underperform in continuous control tasks."...**
> > > > >
> > > > > Thank you for the clarification. You are correct that TD-MPC2 performs extremely well with state-based observations whilst DreamerV3 was designed for image-based observations. This makes a head-to-head comparison difficult and this was a motivation for us to conduct the experiments using discrete latent spaces within a world model for continuous control, in a controlled setting. We (the authors) have discussed your comment and propose to update the abstract as follows (changes are bolded):
> > > > > > In reinforcement learning (RL), world models serve as internal simulators, enabling agents to predict environment dynamics and future outcomes in order to make informed decisions. **While previous approaches leveraging discrete latent spaces, such as DreamerV3, have demonstrated strong performance in discrete action settings and visual control tasks, their comparative performance in state-based continuous control remains underexplored. In contrast, methods with continuous latent spaces, such as TD-MPC2, have shown notable success in state-based continuous control benchmarks.** This paper explores ...
> > > > >
> > > > > Please let us know if you think further changes are required.
> > > > >
> > > > > **AQ2 & AQ4: I disagree with the statement that codebook encodings can ensure a complete ordering. Actually in the example provide, we have $|e(A)-e(B)| = |e(A)-e(C)|$.**
> > > > >
> > > > > We respectfully disagree, as in the general case, codebook encodings can capture ordinal structure. However, you are correct that our example did not exhibit a complete ordering across both dimensions, which is rather misleading. Our apologies for that. We have modified the example so that the ordering is preserved. See below:
> > > > > > It is perfectly plausible that the codebook may encode them as $e(A)=[-0.5,-0.5]$, $e(B)=[0, 0]$, $e(C)=[0.5,0.5]$. In this case, the global ordering ($A < B < C$) is preserved along both dimensions of the codebook. Further to this, it is worth noting that codebook encodings are flexible enough to model ordinal relationships in multiple dimensions. For example, the following code vectors exhibit opposite ordering along their two dimensions $e(D)=[0.5,-0.5]$, $e(E)=[0, 0]$, $e(F)=[-0.5,0.5]$, which adds a level of modelling flexibility.
> > > > >
> > > > > We hope this adequately addresses your concern. Please let us know if anything remains unclear.
> > > > >
> > > > >
> > > > > **AQ5: I still don't understand the literal meaning of "using one-hot and label encodings for the dynamics led to the agent being unable to learn"? Does "unable to learn" mean that the empirical performance is poor?**
> > > > >
> > > > > When experimenting with different discrete encodings, we tried training the dynamics model to use the one-hot and label encodings as input. However, when we replaced the codebook encoding with either one-hot or label encodings, this led to the training curves (environment step vs episode return) flat-lining. That is to say, they never caught the learning signal and were unable to solve the task. If this explanation is clearer, we will update the text with this clarification.
> > > > >
> > > > >
> > > > > **AQ8: Since both TD-MPC2 and DCWM achieve a final success rate of nearly 100%, I recommend plotting the IQM of learning curves (like Fig 4 in TD-MPC2) instead of the final performance. Note that I am not complaining about the performance but solely about the presentation.**
> > > > >
> > > > > We can certainly do this, thank you for the suggestion. We will update these figures to show environment step vs IQM normalized score.
> > > > >
> > > > > **AQ9: Apologies for my earlier unclear comment. My suggestion was specifically regarding FSQ combined with a base method like DreamerV3 (as mentioned in Q10). I am not aiming to add numerous baselines solely for comparison.**
> > > > >
> > > > > Thank you for the clarification. We actually managed to get results for DreamerV3 and TDMPC with FSQ. Please see Figure 17 where we have included results where we replaced DreamerV3's one-hot encoding with FSQ. The results are inconclusive and we believe this to be due to DreamerV3's decoder hurting its performance. For TD-MPC2, please see Figure 21 where we added our discrete (and stochastic) latent space to TD-MPC2. The results for TD-MPC2 indicate that our discrete and stochastic latent space formulation does offer benefits for TD-MPC2, relative to it's continuous and discrete latent space. We think this is an interesting result that motivates further investigation into discrete latent spaces for world models.

---

> > > > > > ### Author Response · Authors · 2024-11-29
> > > > > >
> > > > > > **AQ10 Appendix B.9: The finding that reconstruction harms performance is intriguing. Some preliminary literature [1] has explored the role of the reconstruction component in world models. I believe the weight of the reconstruction loss could play a significant role here...**
> > > > > >
> > > > > > Thank you for the interesting comment and reference. This is a good point and we agree it may have an impact as we did not modify the loss coefficients. We will update this section to make it clear that we did not modify the loss coefficients and suggest this could be what led to the poor performance. We will also cite HarmonyDreamer as a potential solution which warrants further investigation.
> > > > > >
> > > > > > **AQ10 Appendix B.11: I am pleased to see that TD-MPC2 itself benefits from a discrete latent space. These results suggest that the superior performance of DCWM is also influenced by implementation differences compared to TD-MPC2. I recommend highlighting this point more explicitly in future revisions.**
> > > > > >
> > > > > > We agree this is a very interesting observation as it highlights a discrete latent space can benefit TD-MPC2. You are correct that it also highlights that TD-MPC2's other algorithmic differences impact its performance compared DCWM, meaning that a head-to-head comparison does not directly compare the different latent spaces. This is why we include Figures 5 and 12 which test the different latent space designs in a controlled setting. Below we compare how DCWM differs from TD-MPC2 and we will include this in our updated manuscript:
> > > > > >
> > > > > > > 1. TD-MPC2 uses a continuous (but bounded) latent space as it uses simplical normalization (SimNorm). In contrast, DCWM uses a discrete latent space where the latent states are codes from a codebook.
> > > > > > > 2. TD-MPC2 uses a deterministic dynamics model whereas DCWM uses a stochastic dynamics model.
> > > > > > >3. TD-MPC2 uses mean squared error regression for its latent state consistency loss. In contrast, DCWM uses the cross entropy loss.
> > > > > > > 4. TD-MPC2 utilizes SAC as the underlying model-free RL algorithm for learning its policy, whilst DCWM leveraged TD3. As such, TD-MPC2 leverages the maximum entropy RL framework which may provide better exploration.
> > > > > > >5. TD-MPC2 learns its Q-function jointly with the encoder, dynamics, and reward functions. In contrast, DCWM learns the Q-function separately.
> > > > > > >6. TD-MPC2 automatically sets the discount factor $\gamma$ using a heuristic. In contrast, DCWM kept it constant across all tasks.
> > > > > >
> > > > > > We thank the reviewer once again for their detailed and thorough review. We greatly appreciate your feedback and efforts to help us improve our manuscript. Please let us know if anything remains unclear or if you have any further questions or suggestions.

---

> > > > > > > ### Comment · Reviewer_HZL5 · 2024-11-29
> > > > > > >
> > > > > > > The response is perfectly adequate to address any remaining concerns. I believe this paper has been improved significantly through the in-depth discussion. I update my score to 8.

---

### Official Review · Reviewer_hXNL · 2024-11-04

**Soundness:** 3
**Presentation:** 3
**Contribution:** 3
**Rating:** 5
**Confidence:** 4

**Summary:**

This work proposes a model-based RL method for continuous control tasks. Compared with TD-MPC2, the latent space is represented as a discrete codebook. The proposed modifications lead to some improvements over TD-MPC2.

**Strengths:**

1. The paper is well organized and easy to follow.
2. The description of the contribution to the article is simple and concise.
3. The ablation experiment is sufficient and persuasive.

**Weaknesses:**

1. The test for multi-tasking learning is missing. It would be great if the authors could consider extend their studies to multi-task learning and study the differences in codebooks between different control tasks.
2. In some Meta-World tasks, the performance of the proposed method is comparable to TD-MPC2, and the variance is larger.
3. The stochastic transition dynamics in the latent space could be better described and explained.

**Questions:**

1. The authors kept most hyperparameters fixed across all tasks, what parameters will affect the performance of some special tasks?
2. Is the codebook a direct application of quantization-aware training (QAT)?
3. How to determine the weights of the expected code? Is this a hyperparameter of the algorithm?

---

> ### Author Response · Authors · 2024-11-25
>
> We thank the reviewer for their comments and are glad that they found the paper well-written and the presented ablation experiments sufficient.
>
> **W1: The test for multi-tasking learning is missing. It would be great if the authors could consider extend their studies to multi-task learning and study the differences in codebooks between different control tasks.**
>
> **AW1:** We agree that extending DCWM to the multi-task RL setting and learning different codebooks is interesting, however, we intentionally left this for future work as our current paper is focused solely on discrete codebook encodings for world models.
> As we detail in our paper, the use of discrete code embeddings increases the performance of world models. Evaluating the use of a task-dependent variable code embedding, as you propose, is beyond the current scope of this paper especially given the short time window available for running additional experiments in the discussion period.
> However, we appreciate your idea for this possible extension.
>
> **W2: The stochastic transition dynamics in the latent space could be better described and explained.**
>
> **AW2:** We thank the reviewer for bringing this to our attention. In response, we have updated Section 4.1 to include further detail on the stochastic latent transition dynamics. We include our update here for your convenience:
> > As we have a discrete latent space, we formulate the transition dynamics to model the distribution over the next latent state $\mathbf{c}'$ given the previous latent state $\mathbf{c}$ and action $\mathbf{a}$. That is, we model stochastic transition dynamics in the latent space. We denote the probability of the next latent state $\mathbf{c}'$ taking the value of the $i^{\text{th}}$ code $\mathbf{c}^{i}$ as $p_{i} = P\_{\phi}(\mathbf{c}'=\mathbf{c}^{i} \mid \mathbf{c}, \mathbf{a})$. This results in the next latent state following a categorical distribution $\mathbf{c}' \sim \mathrm{Categorical} \left( p_{1},\ldots,p\_{|\mathcal{C}|} \right)$. We use a standard classification setup, where we use an MLP to predict the logits $\mathbf{l} = \{l_{1}, \ldots, l\_{|\mathcal{C}|}\}$. Note that logits are the raw outputs from the final layer of the neural network (NN), which represent the unnormalized probabilities of the next latent state $\mathbf{c}'$ taking the value of each discrete code in the codebook $\mathcal{C}$. The logit for the $i^{th}$ code is given by $l_{i} = d_{\phi,i} (\mathbf{c}, \mathbf{a}) \in \mathbb{R}$.
> We then apply the softmax operation to obtain the probabilities $p_{i}$ of the next latent state taking each discrete code in the codebook $\mathcal{C}$, i.e., $p\_{i} = \text{softmax}\_{i}(\mathbf{l})$. DCWM uses the discrete codes $\mathbf{c}$ as its latent state to make dynamics, reward $R\_{\xi}(\mathbf{c},\mathbf{a})$,
> value $\mathbf{q}\_{\psi}(\mathbf{c},\mathbf{a})$, and policy $\pi\_{\eta}(\mathbf{c})$ predictions. Our hypothesis is that learning these components with a discrete latent space will be more efficient than with a continuous
> latent space.
>
>
> Please let us know if we have adequately improved our explanation and if you have any further suggestions to help us improve it.
>
> **Q1: The authors kept most hyperparameters fixed across all tasks, what parameters will affect the performance of some special tasks?**
>
> **AQ1:** Due to limited computing resources we did not fine-tune hyperparameters for each task, but instead aimed to use the same set of hyperparameters across all tasks, similar to prior work on model-based RL with world models. However, in more complex tasks where training takes longer, e.g., the Dog and Humanoid tasks, noise scheduling affects the training. Therefore, we reduced the exploration noise later.
>
> **Q2: Is the codebook a direct application of quantization-aware training (QAT)?**
>
> **AQ2:** While there are possible parallels between embedding-discretization and quantization-aware training, the latter is usually concerned with robustifying neural networks to quantization effects at the deployment stage. Hence, we would argue that both approaches differ both in their application and desired outcomes.
>
> **Q3: How to determine the weights of the expected code? Is this a hyperparameter of the algorithm?**
>
> **AQ3:** We would like to ask the reviewer to clarify this question if possible. We are unsure which weights you refer to. As a clarification, we do not discretize the weights of the neural network, but we discretize the embedding space - this is a different approach to quantization-aware training.

---

> > ### Comment · Reviewer_hXNL · 2024-11-26
> >
> > I am not talking about QAT in Q3. The authors mentioned at the end of Section 4 that, "we take the expected code, which is a weighted sum over the codes in the codebook." How to determine the weights when summing over the codes?

---

> > > ### Author Response · Authors · 2024-11-26
> > >
> > > Thank you for the quick response!
> > >
> > > DCWM's latent space dynamics model resembles a neural network (NN) classifier. As such, the NN directly predicts the logits $\mathbf{l} = \\{l_{1}, \ldots, l_{|\mathcal{C}|}\\}$, that is, the unnormalized probabilities of the next latent code $\mathbf{c}'$ taking the value of each code in the codebook $\mathbf{c}^{i} \in \mathcal{C}$, given by $p_{i} = P_{\phi}(\mathbf{c}'=c^{i} \mid \mathbf{c}, \mathbf{a})$. We obtain the probabilities (aka weights) using softmax $p_{i} = \text{softmax}_{i}(\mathbf{l})$. At planning time, these probabilities are the "weights when summing over the codes?".
> > >
> > > We refer the reviewer to Lines 192-201 in the revised paper where we have improved our discussion of DCWM's latent space dynamics (included here for convenience):
> > > > As we have a discrete latent space, we formulate the transition dynamics to model the distribution over
> > > the next latent state $\mathbf{c}'$ given the previous latent state $\mathbf{c}$ and action $\mathbf{a}$.
> > > That is, we model stochastic transition dynamics in the latent space.
> > > We denote the probability of the next latent state $\mathbf{c}'$ taking the
> > > value of the $i^{\text{th}}$ code $\mathbf{c}^{i}$ as $p_{i} = P_{\phi}(\mathbf{c}'=c^{i} \mid \mathbf{c}, \mathbf{a})$.
> > > This results in the next latent state following a categorical distribution $\mathbf{c}' \sim \mathrm{Categorical} \left( p_{1},\ldots,p_{|\mathcal{C}|} \right)$.
> > > We use a standard classification setup, where
> > > we use an MLP to predict the logits $\mathbf{l} = \\{l_{1}, \ldots, l_{|\mathcal{C}|}\\}$.
> > > Note that logits are the raw outputs from the final layer of the neural network (NN), which represent the unnormalized probabilities
> > > of the next latent state $\mathbf{c}'$ taking the value of each discrete code in the codebook $\mathcal{C}$.
> > > The logit for the $i^{th}$ code is given by $l_{i} = d_{\phi,i} (\mathbf{c}, \mathbf{a}) \in \mathbb{R}$.
> > > We then apply the softmax operation to obtain the probabilities $p_{i}$ of the next latent state taking each
> > > discrete code in the codebook $\mathcal{C}$, i.e., $p_{i} = \text{softmax}_{i}(\mathbf{l})$.
> > >
> > > We hope this new text will help avoid this confusion in the future. Please let us know if anything remains unclear or if you have any suggestions to help us improve it.

---

> > > ### Author Response · Authors · 2024-11-29
> > >
> > > The discussion period will be coming to an end soon and we would greatly appreciate it if you could let us know if we have adequately addressed your questions. Further to this, we have made significant improvements based on the feedback raised by all reviewers, specifically:
> > >
> > > - **Improved clarity** Based on your feedback (thanks for pointing this out), we have improved the description of our latent stochastic transition dynamics model. Please see Lines 193-204.
> > > - **Shown that DCWM improves TD-MPC2** In Figure 21, we have shown that replacing TD-MPC2's continuous latent space with our discrete and stochastic latent space improves performance. This is an interesting result that highlights our findings are transferrable outside of our specific setup.
> > > - **Improved empirical study** We have increased the number of Meta-World tasks that we evaluate DCWM on from 7 to 45, significantly improving the statistical significance of our empirical study. Please see Figures 3 and 14.
> > > - **Comparison of our FSQ vs VQ-VAE** In Figure 15, we have shown the impact of using vector quantization (VQ) in place of our finite scalar quantization (FSQ). The results indicate that the VQ-VAE approach requires environment-specific adjustments which undermine its general applicability compared to our approach of using FSQ.
> > >
> > > These changes have substantially strengthened our paper's contributions. We'd also like to draw your attention to the extensive discussion with other reviewers who found these improvements positive and raised their scores. We believe these enhancements strengthen our paper significantly and we'd greatly appreciate it if you could reassess your score in light of these updates. We're committed to further improving our work and would be grateful for any additional feedback you may have. Once again, we'd like to thank you for your time and efforts.

---

### Official Review · Reviewer_Nctn · 2024-11-04

**Soundness:** 4
**Presentation:** 3
**Contribution:** 3
**Rating:** 8
**Confidence:** 3

**Summary:**

The authors contribute a novel method, called DCWM, for model-based RL on continuous control tasks. Specifically, building on the works of Dreamer V3 and TD-MPC2 (prior model-based continuous control methods), the paper sheds light on the impact of how these methods encode latent state (different ways of discrete encoding vs. continuous), how that state is trained (a categorical loss or a regressive loss), and the impact of stochasticity in modeling dynamics. The paper conducts experiments and ablations on Deepmind Control Suite and Meta World to identify which components are most impactful on performance, and combines them to present the aforementioned method, while comparing to baselines from literature. The authors conclude that learning a discrete latent state (specifically one encoded by codebook quantization over other methods), and training with a cross entropy objective results in improved performance.

**Strengths:**

The authors present a strong paper, with excellent methodology and solid experimental results. They compare with two SOTA baselines, Dreamer V3 and TD-MPC2, and demonstrate that their method indeed outperforms or matches the baseline performance. They conduct additional experiments (sections 4.3, 4.4 and 4.5) to understand how different variables affect performance, which enhances the paper's contributions to bettering understanding latent spaces in world model learning. The paper's clarity is also very good, with a great presentation of background and method. The authors provide sufficient detail, such as the discussion around the fixed FSQ codebook illustration in Figure 2 or the extensive documentation of various tricks used to improve performance (lines 309, 352, 355, etc.). Additionally, the supplementary material is strong, with detailed extra information on method, environment, architecture, etc, and addition ablation experiments. The results contributed are significant, not only advancing the state of the art for model-based continuous control (to my knowledge), but also contributing to the field's understanding of the impacts of latent space encodings, losses, and stochasticity in modeling transition dynamics. I would argue that the method merits originality as well, since it cleverly combines prior work (the consistency approach from TD-MPC2 and discrete encodings/cross entropy objective from Dreamer V3) and introduces novel elements as well (codebook quantization).

Overall, this is a strong contribution to the field, and I recommend acceptance.

**Weaknesses:**

The paper is free of major flaws: it has thoroughly conducted experiments and is clear and well-written. However, I have identified some nits and clarifying questions listed in the section below that would be minor improvements.

**Questions:**

Question:
- I imagine that taking the expected code might sometimes result in an invalid state, as discussed here:

>  Whilst the expected value of
a discrete variable does not necessarily take a valid discrete value, we find it effective in our setting.

Is that simply then fed into the dynamics model (after normalization etc.) and the reward model MLPs?
- Relatedly, when you normalize the code to [-1, 1], you claim it improves performance. Is there a reference for this or was this empirically observed? Would you be able to speak to why this is required? (One thought I had was it normalizes the different quantization levels $L_i$, i.e. the two channels would otherwise have values ranging from [-5, 5] and [-3, 3]).

Nits (not affecting score):
- line 217, formatting for `i-th` and `d-th` is inconsistent, and should use latex
- line 352 typo in 'warms starts'
- line 465, missing 'and' before models in 'entropy), **and** models'

---

> ### Comment · Reviewer_Nctn · 2024-11-25
>
> Keeping my score, as the authors have not provided any additional response.

---

> ### Author Response · Authors · 2024-11-25
>
> We are happy to see that you found our paper excellent, insightful and in a good state for publication. Please find below our clarifications to your questions:
>
> **Q1: I imagine that taking the expected code might sometimes result in an invalid state, as discussed here: "Whilst the expected value of a discrete variable does not necessarily take a valid discrete value, we find it effective in our setting." Is that simply then fed into the dynamics model (after normalization etc.) and the reward model MLPs?**
>
> **AQ1:** Yes you are correct, we simply feed the expected code as the input to the dynamics, reward, Q-function and prior policy.
>
> **Q2: Relatedly, when you normalize the code to [-1, 1], you claim it improves performance. Is there a reference for this or was this empirically observed? Would you be able to speak to why this is required? (One thought I had was it normalizes the different quantization levels, i.e. the two channels would otherwise have values ranging from [-5, 5] and [-3, 3]).**
>
> **AQ2:**  The insight that normalization of the code to the range of [-1, 1] improves performance was indeed raised in Mentzer et al. (2023). We will add a mention of this in the new version of the paper.
>
> Please also see our general response (and revised draft) where we have provided new experiments to strengthen our submission.

---

### Official Review · Reviewer_BJj9 · 2024-11-05

**Soundness:** 3
**Presentation:** 3
**Contribution:** 2
**Rating:** 8
**Confidence:** 3

**Summary:**

The authors propose a new architecture for world modeling for continuous control that utilizes discrete representations. Rather than use a learned codebook or one-hot encodings, they utilize finite scalar quantization (FSQ) which simplifies the loss function and stabilizes the learning according to their claims. They build upon the TD-MPC2 algorithm which utilizes model predictive path integral (MPPI) to search the world model for high value actions.

They test their algorithm in a number of high dimensional continuous control tasks including Meta-World manipulation and DeepMind Dog and Humanoid tasks, showing that their algorithm achieves higher success and learns with fewer samples compared to baselines. In addition to comparison against baselines, they also provide experiments supporting their choice of discrete encodings, showing that they can achieve similar performance to one-hot encodings while increasing training efficiency.

**Strengths:**

The paper is well-written and explains all necessary background material in sufficient detail to understand the algorithm. While none of the individual components are novel, the experimental evaluation provides evidence that the proposed algorithmic details are critical for the success of model-based learning.

**Weaknesses:**

The one question that I have relates to the use of REDQ to reduce bias in the TD learning. Since the baseline does not seem to use this method, it begs the question of how much of the demonstrated performance is due to this component. It would be nice to see how DCWM compares to a version that uses the standard one or two Q functions to make sure the performance gap is in fact due to the choice of latent spaces.

**Questions:**

What does the algorithm performance look like without using REDQ?

What are the computational costs or benefits to this method over TD-MPC in terms of runtime?

---

> ### Author Response · Authors · 2024-11-25
>
> We thank Reviewer BJj9 for their valuable feedback. We are glad they found the paper well-written and that our experimental evaluation provides evidence that the proposed algorithmic details are critical to its success.
>
> **W1: The one question that I have relates to the use of REDQ to reduce bias in the TD learning. Since the baseline does not seem to use this method, it begs the question of how much of the demonstrated performance is due to this component. It would be nice to see how DCWM compares to a version that uses the standard one or two Q functions to make sure the performance gap is in fact due to the choice of latent spaces.**
>
> **Q1: What does the algorithm performance look like without using REDQ?**
>
> **AW1:** First of all, let us highlight that we did not use REDQ to obtain a performance improvement over TD-MPC2. On the contrary, we intentionally used the same strategy for the critic as was used in the TD-MPC2 paper. They used an ensemble of five critics and sub-sample two critics before calculating either the mean or the minimum of the two sub-sampled Q-values. So once again, let us clarify that we use the same critic strategy as the TD-MPC2 baseline. Nevertheless, to avoid confusion in the future, we have updated Line 293 to explicitly state that this was done in TD-MPC2:
> >"We also reduce bias in the TD target by following REDQ (Chen et al., 2021) and learning an ensemble of $N_{q}=5$ critics, as was done in TD-MPC2."
>
> The reviewer is right that results using the standard double Q learning would provide further insights into what aspects of our method contribute to its performance. As such, we have run experiments using the standard double Q learning setup and report results in the Reacher Hard, Walker Walk, Dog Run and Humanoid Walk tasks. This experiment provides further insights into what parts of our algorithm contribute to its high performance, so we have added the figure to Appendix B.8. We thank the reviewer for bringing this up and hope this answers the question.
>
>
> **Q2: What are the computational costs or benefits to this method over TD-MPC in terms of runtime?**
>
> **AQ2:** We thank the reviewer for the good question. Our method (DCWM) has a similar runtime to TD-MPC2 when using our default hyperparameters. For example, in the Dog Run task, TD-MPC2 takes roughly 22.5 hours to reach 2M time steps while our method (DCWM) takes 28.5 hours. We will update the paper to include this.
>
>
> We thank Reviewer BJj9 for their feedback. Please let us know if you have any further questions or if you have any suggestions that can help us improve our paper's score.

---

> > ### Comment · Reviewer_BJj9 · 2024-12-01
> >
> > Thank you for your response! Based on your clarifications I have changed my score

---

### Official Review · Reviewer_RvJB · 2024-11-08

**Soundness:** 3
**Presentation:** 3
**Contribution:** 3
**Rating:** 5
**Confidence:** 2

**Summary:**

This paper explores the use of discrete latent spaces for continuous control with world models. The authors introduce DCWM: Discrete Codebook World Model, a model-based RL method which surpasses recent state-of-the-art algorithms. it’s a well written paper in general.

**Strengths:**

1. The authors demonstrate that quantized discrete codebook encodings are more effective representations for continuous control, compared to alternative encodings, such as one-hot and label-based encodings.

2. The authors  introduce DCWM: Discrete Codebook World Model, a model-based RL method which surpasses recent state-of-the-art algorithms, including TD-MPC2 and DreamerV3, on continuous control benchmarks.

**Weaknesses:**

What it misses I think  is  a comparison of the method with other approaches that are not based on the same embeddings ideas. For example I think MAMBA and Hungry hungry hippos (H3) apply to similar scenarios.

**Questions:**

I think a comparison of the method with other approaches that are not based on the same embeddings ideas can be added.

---

> ### Author Response · Authors · 2024-11-25
>
> We are glad to see that you found the paper well-written and that you appreciated our research into the use of discrete latent spaces for model-based reinforcement learning with world models.
>
> **Q1: I think a comparison of the method with other approaches that are not based on the same embeddings ideas can be added.**
>
> **W1: What it misses I think is a comparison of the method with other approaches that are not based on the same embeddings ideas. For example, I think MAMBA and Hungry hungry hippos (H3) apply to similar scenarios.**
>
> **AW1:** We want to highlight that we indeed consider, in our experiments and ablation studies, different baselines using different embedding spaces from continuous (TD-MPC2) to the use of discrete one-hot encodings used in DreamerV3. For comparisons between model-based reinforcement learning methods utilizing latent embeddings for planning/rollouts and RL methods not using latent embeddings we would like to refer you to the studies performed in e.g. [Hansen et al. 2023] and [Hafner et al., 2023]. Please, also see our ablation of different encoding styles in the latent embedding in Figure 7.
>
> Lastly, we are unsure how a comparison of language-modelling methods such as Hungry hungry hippos (H3), which is used to encode text sequences in LLMs, would benefit this paper which tackles continuous control tasks in reinforcement learning settings. Extending the presented approach to language tasks appears to us to be out-of-scope and a serious extension of the presented approach, warranting a new paper. Therefore, we would like to invite the reviewer to motivate or clarify this comparison in a follow-up if possible to help us understand the benefits of adding H3.

---

> ### Author Response · Authors · 2024-11-28
>
> Hi, we noticed you adjusted your score. Can you elaborate on why you decreased the score? We would love the opportunity to improve our paper based on your feedback.

---

> ### Author Response · Authors · 2024-11-29
>
> The discussion period will be coming to an end soon and we would greatly appreciate your feedback. We have made significant improvements based on the feedback raised by all reviewers, specifically:
>
> - **Improved empirical study evaluating different embeddings (Figure 5)** We have increased the number of tasks in our comparison of different latent space embeddings (Figure 5) from 4 tasks to 20 tasks (10 DMControl and 10 Meta-World). This significantly improves the statistical significance of our empirical study where we compare the impact of different latent space embeddings. The results show that DCWM's discrete latent space combined with using Gumbel-softmax sampling when making multi-step dynamics predictions, offers improved sample efficiency over continuous latent spaces.
> - **Shown that DCWM improves TD-MPC2** In Figure 21, we have shown that replacing TD-MPC2's continuous latent space with our discrete and stochastic latent space improves performance. This is an interesting result that highlights our findings are transferrable outside of our specific setup.
> - **Improved empirical study** We have increased the number of Meta-World tasks that we evaluate DCWM on from 7 to 45, significantly improving the statistical significance of our empirical study. Please see Figures 3 and 14.
> - **Comparison of our FSQ vs VQ-VAE** In Figure 15, we have shown the impact of using vector quantization (VQ) in place of our finite scalar quantization (FSQ). The results indicate that the VQ-VAE approach requires environment-specific adjustments which undermine its general applicability compared to our approach of using FSQ.
>
> These changes have substantially strengthened our paper's contributions. We'd like to draw your attention to our in depth discussion with Reviewer HZL5, who initially gave our paper a lower score but has since increased after seeing these improvements. We believe these enhancements strengthen our paper significantly and we'd greatly appreciate it if you could reassess your score in light of these updates. We would be grateful for any additional feedback you may have, and once again, we'd like to thank you for your time and efforts.

---

### Author Response · Authors · 2024-11-25
**Response to all reviewers**

We thank all reviewers for their valuable feedback, which helped us to improve our paper even further. We are glad that all reviewers found the paper well written and that reviewers Nctn, HZL5, and hXNL agree our investigation into latent space design is especially valuable for the model-based RL community. In this response to all reviewers, we detail the extra experiments we have included in light of the reviewer's comments.

## Experiments
1. **More tasks & aggregate metrics** In response to Reviewer HZL5, we have improved our empirical study by increasing the number of tasks and reporting aggregate metrics across all tasks.
    - **More Meta-World tasks** We have evaluated DCWM in 45 Meta-World tasks. This increases the number of MetaWorld tasks from 7 to 45, which is a significant increase in the number of tasks. See Figure 12 for the training curves for all 45 Meta-World tasks.
    - **MyoSuite Musculoskeletal results (App. B.10)**  In response to HZL5, we have evaluated DCWM in five musculoskeletal tasks from MyoSuite. We report the individual training curves in Figure 18 and we report aggregate metrics in Figure 17. On average, DCWM performs well, generally matching TD-MPC2 and outperforming the other baselines (DreamerV3, SAC, TD-MPC). We also observe that DCWM also has more consistent asymptotic performance, suggesting that it is more stable than TD-MPC2.
 These results demonstrate that DCWM is applicable in a wide range of environments and strengthens our paper's empirical study.
    - **Aggregate performance metrics from Rliable [1]** We now report aggregate metrics (median, interquartile mean (IQM), mean, optimality gap) with $95\\%$ stratified confidence intervals over the 29 DMControl tasks (see Figure 3) and the 45 Meta-World tasks (see Figure 5). This is a much more robust and efficient approach to reporting results and we believe this significantly boosts our empirical study.
2. **DCWM with VQ-VAE (App. B.7)** To understand how the choice of using FSQ for discretization contributes to the performance of our algorithm, we replaced the FSQ layer with a standard Vector Quantization layer. We evaluated the methods in Walker Walk, Dog Run, Humanoid Walk, and Reacher Hard. We used standard hyperparameters, $\beta=0.25$, and an EMA-updated codebook with a size of $256$ and either $256$ (dog) or $128$ (other tasks) channels per dimension. We did not change other hyperparameters from DCWM. However, we found that to approach the performance of standard FSQ, VQ-VAE needs environment-dependent adjusting of the planning procedure. In Humanoid Walk, the performance of FSQ aligns closely with the VQ-VAE with a weighted sum over the codes in the codebook for planning (expected code) but significantly outperforms sampled VQ-VAE. Conversely, standard sampling is superior in Reacher Hard, which is unsurprising, as the discrete codes in VQ-VAE do not have an inherent order like in FSQ. The necessary environment-specific adjustments for VQ-VAE undermine its general applicability compared to FSQ.
3. **Reconstruction prevents DreamerV3 performing well (App. B.9)** We have added a section comparing DCWM and DreamerV3. Figure 15 shows that replacing DreamerV3's one-hot encoding (orange) with DCWM's codebook encoding (blue) offers no improvement. However, we also see that DreamerV3 is unable to solve the difficult Dog Run and Humanoid Walk tasks in the 2M environment steps, and these tasks are where the codebook encodings particularly shine. We hypothesize that DreamerV3's poor performance in the Dog Run and Humanoid Walk tasks results from its decoder struggling. We provide evidence for this in Figure 16, where we add a reconstruction term to DCWM's world model loss. The results show that in no environments does reconstruction aid learning, and in some tasks, such as the difficult Dog Run and Humanoid Walk tasks, including the reconstruction term has a significant detrimental effect on the performance, and can even prevent learning completely. These new experiments provide further insights which help us understand the performance gap between DreamerV3 and DCWM.
5. **Ablation of REDQ critic vs standard double Q (App. B.8)** In response to BJj9, we have included experiments showing how DCWM performs when replacing our ensemble of critics with the standard double Q approach. We report results in Walker Walk, Dog Run, Humanoid Walk, and Reacher Hard. Please see App. B.8 where we have written up the results. The results highlight that the ensemble critic approach used by DCWM and TD-MPC2 does improve performance over the standard double Q approach. Note that we intentionally used the same ensemble critic approach as TD-MPC2 so that our changes to the latent space were isolated from other design choices such as the critic.

## References
[1] - Rishabh Agarwal, Max Schwarzer, Pablo Samuel Castro, Aaron C Courville, and Marc Bellemare. Deep Reinforcement Learning at the Edge of the Statistical Precipice. NeurIPs 2021.

---

> ### Author Response · Authors · 2024-11-28
> **Follow up to all reviewers**
>
> We thank the reviewers for their continued input. We have been working hard to incorporate your constructive feedback and we have now updated and improved our manuscript once again. Below we summarize the improvements:
>
> - **How does TD-MPC2 perform with DCWM's latent space? (Appendix B.11)** Reviewer HZL5 highlighted (and Reviewer Nctn agreed) that the best way to evaluate our quantized latent space was to directly replace the latent space in TD-MPC2 for a head-to-head comparison. We have now evaluated TD-MPC2 with DCWM's latent space in 10 DMControl and 10 Meta-World tasks. The results show that using DCWM inside TD-MPC2 improves sample efficiency in the 10 DMControl tasks, whilst in the 10 Meta-World tasks, the performance of all methods seems about equal. This suggests that discrete stochastic latent spaces can be advantageous in the context of world models for continuous control. This is an interesting result that we believe motivates further research into discrete and stochastic latent spaces for world models. We are happy with the new results and we thank Reviewer HZL5 for the suggestion. Please see Appendix B.11 and Figure 21 for further details and the full write-up. If reviewers agree, we are happy to move Figure 21 into the main text. However, for the ease of the rebuttal, we thought it would be easiest to add a new appendix solely dedicated to the new results.
> - **Increased number of tasks for ablations (Figure 5)** We have increased the number of tasks in the latent space comparison (Figure 5) from 4 tasks to 20 tasks (10 DMControl and 10 Meta-World). This significantly improves the statistical significance of our empirical study where we compare the impact of different latent space designs. The results show that DCWM's discrete latent space combined with using Gumbel-softmax sampling when making multi-step dynamics predictions, offers improved sample efficiency over continuous latent spaces. Perhaps surprisingly, continuous stochastic latent spaces do not offer the same benefit. Please see Section 5.2 and Figure 5 for further details.
> - **Combined and moved figures** To make space for these new results, we moved the DMControl and Meta-World aggregate performance metrics plots to Appendix B.1 and have added a figure in the main text to summarize the aggregate metrics for both DMControl and Meta-World (Figure 3).
>
> Once again, we would like to thank all the reviewers for their time. Please let us know if there are any remaining questions or concerns. We would be happy to run any additional experiments that you think would improve our paper.

---

### Meta-Review · Area_Chair_nUZ1 · 2024-12-19

**Metareview:**

This paper introduces DCWM, exploring discrete codebook encodings for continuous control in reinforcement learning. The reviewers found the investigation into latent space design valuable and the paper well-written. Through extensive revisions, the authors significantly expanded their empirical evaluation across 45 Meta-World and 5 MyoSuite tasks, providing comprehensive ablation studies including direct comparisons with TD-MPC2's latent space. Given the paper's thorough evaluation and meaningful contributions to model-based RL, particularly in demonstrating the advantages of discrete stochastic latent spaces, I recommend acceptance.

**Additional Comments On Reviewer Discussion:**

While initial concerns about discrete latent space claims and experimental breadth existed, these were effectively addressed through additional experiments.

---

### Decision · Program_Chairs · 2025-01-22

Accept (Poster)